# Reference Soil Groups Map of Ethiopia Based on Legacy Data and Machine Learning Technique: EthioSoilGrids 1.0

Ashenafi Ali[1,2,3,4], Teklu Erkossa[3], Kiflu Gudeta[2], Wuletawu Abera[4], Ephrem Mesfin[2], Terefe Mekete[2], Mitiku Haile[6], Wondwosen Haile[7], Assefa Abegaz[1], Demeke Tafesse[12], Gebeyhu Belay[7], Mekonen Getahun[8,9], Sheleme Beyene[10], Mohamed Assen[1], Alemayehu Regassa[11], Yihenew G. Selassie[9], Solomon Tadesse[12], Dawit Abebe[13], Yitbarek Wolde[13], Nesru Hussien[2], Abebe Yirdaw[2], Addisu Mera[2], Tesema Admas[2], Feyera Wakoya[2], Awgachew Legesse[2], Nigat Tessema[2,10], Ayele Abebe[14], Simret Gebremariam[2], Yismaw Aregaw[2], Bizuayehu Abebaw[2], Damtew Bekele[12], Eylachew Zewdie[7], Steffen Schulz[3], Lulseged Tamene[4], and Eyasu Elias[2,5]

[1]Department of Geography and Environmental Studies, Addis Ababa University (AAU), Addis Ababa, Ethiopia

[2]Ministry of Agriculture (MoA), Addis Ababa, Ethiopia

[3]Deutsche Gesellschaft für Internationale Zusammenarbeit (GIZ), Ethiopia

[4]International Centre for Tropical Agriculture (CIAT), Addis Ababa, Ethiopia

[5]Centre for Environmental Science, Addis Ababa University, Ethiopia

[6]Mekelle University, Mekelle, Ethiopia

[7]Private Consultant, Addis Ababa, Ethiopia

[8]Amhara Design and Supervision Enterprise (ADSE), Bahir Dar, Ethiopia

[9]BahirDar University (BDU), Bahir Dar, Ethiopia

[10]Hawassa University (HU), Hawassa, Ethiopia

[11]Jimma University (JU), Jimma, Ethiopia

[12]Ethiopian Construction Design and Supervision Works Corporation (ECDSWCo), Addis Ababa, Ethiopia

[13]Engineering Corporation of Oromia, Addis Ababa, Ethiopia

[14]National Soil Testing Centre, MoA, Addis Ababa, Ethiopia.

**Correspondence**: Ashenafi Ali (ashenafi.ali@aau.edu.et)

**Abstract.** Up-to-date digital soil resource information and its comprehensive understanding are crucial to supporting crop production and sustainable agricultural development. Generating such information through conventional approaches consumes time and resources, and is difficult for developing countries. In Ethiopia, the soil resource map that was in use is qualitative, dated (since 1984), and small-scaled (1:2 M) which limit its practical applicability. Yet, a large legacy soil profile data accumulated over time and the emerging machine learning modelling approaches can help in generating a high-quality quantitative digital soil map that can provide better soil information. Thus,

a group of researchers formed a coalition of the willing for soil and agronomy data sharing and collated about 20,000 soil profile data and stored them in a central database. The data were cleaned and harmonised using the latest soil profile data template and 14,681 profile data were prepared for modelling. Random Forest was used to develop a continuous quantitative digital map of 18 World Reference Base (WRB) soil groups at 250 m resolution by integrating environmental covariates representing major soil-forming factors. The map was validated by experts through a rigorous process involving senior soil specialists/pedologists checking the map based on purposely-selected district level geographic windows across Ethiopia. The map is expected to have tremendous value in soil management and other land-based development planning, given its improved spatial resolution and quantitative digital representation.

**Keywords**: soil profiles, environmental covariates, modelling, expert validation, Reference Soil Group

# 1    Introduction

Soils are important resources that support the development and production of various economic, social, and ecosystem services, and are useful in climate change mitigation and adaptation (Baveye et al., 2016). Data on soils' physical and chemical characteristics and their spatial distribution are needed to define and plan their functions over time and space, which are important steps towards sustainable use and management of soils (Elias, 2016; Hengl et al., 2017).

 In Ethiopia, soil surveys and mapping have been conducted at various scales with varying scopes, approaches, methodologies, qualities, and levels of detail (Abayneh, 2001; Abayneh and Berhanu, 2007; Berhanu, 1994; Elias, 2016; Zewdie, 2013). The most recent country-wide digital soil mapping efforts focused primarily on soil characteristics (Ali et al., 2020; Iticha and Chalsissa, 2019; Tamene et al., 2017), although soil class maps are equally important for allocating a particular soil unit for specific use (Leenaars et al., 2020a; Wadoux et al., 2020). Many attempts have been made to improve digital soil information systems (Hengl et al., 2021, 2017, 2015; Poggio et al., 2020). However, the initiatives were based on limited and unevenly distributed soil profile data (e.g., 1.15 soil profiles per 1,000 km$^2$ for Ethiopia) which restricts the accuracy and applicability of the products.

In Ethiopia, thousands of soil profile data have been collected since the 1960s (Erkossa et al., 2022), but these data were scattered across different institutions and individuals (Ali et al., 2020). Furthermore, country-wide quantitative and gridded spatial soil type information does not exist (Elias, 2016). The Ethiopian Soil Information System (EthioSIS) project attempted to develop a countrywide

digital soil map focusing on topsoil characteristics, including plant nutrient content, but overlooked soil resource mapping (Ali et al., 2020; Elias, 2016), despite a strong need for a high-resolution soil resource map (Mulualem et al., 2018).

Ethiopia has an area of about 1.14 mill. km$^2$ consisting of varied environments, making its soils extremely heterogeneous. Capturing the heterogeneity using conventional soil survey and mapping approaches is an expensive and time-consuming endeavour (Hounkpatin et al., 2018). This can be circumvented using available legacy soil profile data accumulated over decades  and tapping into the potential of  advanced analytical techniques to develop high-resolution digital soil maps (Hounkpatin et al., 2018; Kempen, 2012, 2009).Therefore, the objectives of this study were to (1) develop a national legacy soil profile dataset that can be used as an input for various digital soil mapping exercises, and (2) generate an improved 250 m digital Reference Soil Groups (RSGs) map of Ethiopia.

## 2  Methods

### 2.1 The study area

The study area covered the entire area of Ethiopia (1.14 mill. km$^2$) located between 3°N and 15° N, and between 33° E and 48° E (Figure 1). The topography of the country is marked by a large altitudinal variation, ranging from 126 meter below sea level at Dalol in the northeast to 4,620 m  at Ras Dashen Mountain in the northwest (Billi, 2015; Enyew and Steeneveld, 2014). Ethiopia's wide range of topography, climate, parent material, and land use types created conditions for the formation of different soil types (Abayneh, 2005; Berhanu and Ochtman, 1974; Donahue, 1972; Mesfin, 1998; Nyssen et al., 2019; Virgo and Munro, 1978; Zewdie, 2013, 1999). More than 33% of the country is covered by the central, upper and highland complex (Abegaz et al., 2022), which embraces Africa's most prominent mountain system (Hurni, 1998).

85

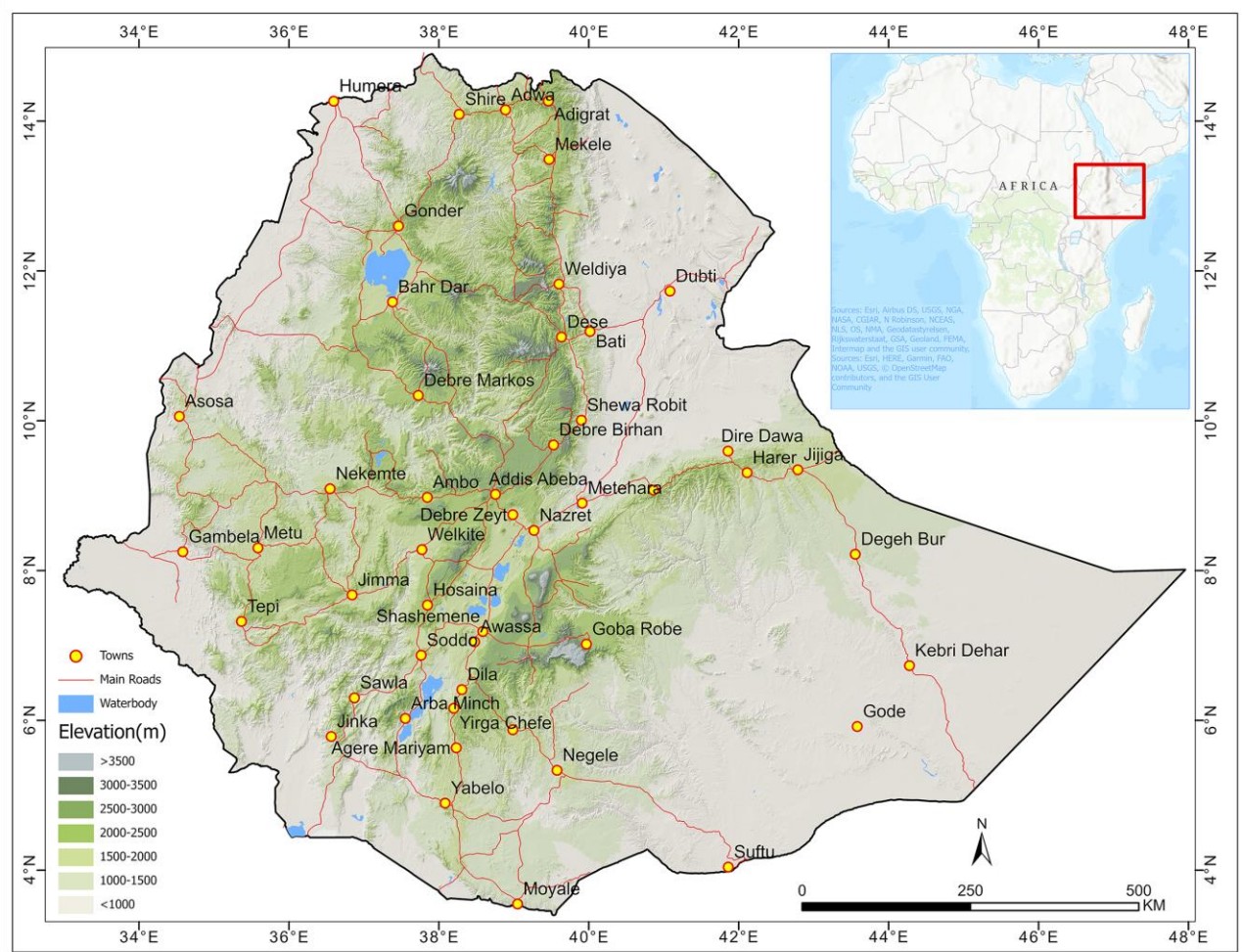

**Figure 1.** Location map of Ethiopia, overview map © Esri World Topographic Map.

The country's complex topography strongly determines both rainfall and temperature patterns, by modifying the influence of the large-scale ocean-land-atmosphere pattern, thus creating diverse localised climates. Spatially, rainfall is characterised by a general decreasing trend in the direction from the west- to east, north, northeast, south and southeast. The lowlands in the southeast and northeast, covering approximately 55% of the country's land area, are characterised by arid and semi-arid climates. Annual rainfall ranges from less than 300 mm in the south-eastern and north-western lowlands to over 2,000 mm in the southwestern (southern portion of the western highlands). The eastern lowlands get rain twice a year, in April–May and October–November, with two dry periods in between. The total annual precipitation in this region varies from less than 500 to 1,000 mm. The driest of all regions is the Denakil Plain, which receives less than 500 mm and sometimes none (Fazzini et

al., 2015). Temperatures are also greatly influenced by the rapidly changing altitude and the mean monthly values vary from ~35°C in the northeast lowlands to less than 7.5°C over the north and central highlands.

The country is characterised by a wide variety of geological formations (Abyneh, 2005; Alemayehu et al., 2014; Elias., 2016; Jarvis et al., 2011; Zewdie, 2013). These include (i) recent and old volcanic activities; (ii) the highlands consisting of igneous rocks (mainly basalts); (iii) steep-sided valleys characterised by strong colluvial and alluvial deposits; (iv) metamorphic rocks exposed by denudation process; and (v) various sedimentary rocks like limestone and sandstone in the relatively lower areas.

Diverse biophysical factors affecting the spatial distribution of vegetated land cover which in turn both as single and combined factors result in diverse soil types and properties across Ethiopia's landscapes (Hurni, 1998; Nyssen et al., 2019; WLRC, 2018). The spatio-temporal vegetation cover of the country has been characterised by a long history of landuse-landcover changes(WLRC, 2018). In terms of the type and spatial coverage of major landuse/landcover classes, woody vegetation (forest, woodland, and shrub and bush lands) covers about 57% of the country in accordance with the national 2016 map (WLRC, 2018). This is followed by cultivated land (20%) and grasslands (12%). Barren lands are estimated to cover about one-tenth of the area of the country while other minor lands with ecological significance (i.e., wetlands, water bodies and sub-afro-alpine and afro-alpine ) cover about 1.2% of the country's land mass.

## 2.2 Legacy soil profile data collation and preparation

The soil profile data generated over decades through various soil survey missions were kept in a variety of formats with limited accessibility. There has been no institution with a mandate to coordinate the generation, collation, harmonisation, and sharing of soil profile data. This led to the formation of a group of individuals and institutions who were willing to exchange soil and agronomy data. Established in 2018, the group known as the Coalition of the Willing (CoW) was committed to addressing the challenges posed by the lack of the soil and agronomy data access and sharing in the country (Tamene et al., 2021).

The CoW conducted a national soil and agronomy data ecosystem mapping which revealed that a plethora of legacy soil resource data sets do exist across different institutions and individuals (Ali et al., 2020). The assessment also revealed that a sizable proportion of the data holders were willing to

share the data in their custody, provided that some regulations are put in place to administer the data.
The CoW developed and approved internal data sharing guidelines (CoW, 2020), and facilitated data
collation campaigns, which involved both formal and informal approaches to data holders.
Through a data collation campaign, soil profile data collected between the 1970s and 2021 were
acquired from over 88 diverse sources (Ali et al., 2020; Tamene et al., 2021). Initially, 8,000 profile
data points were collated and subjected to improved modelling techniques to create a provisional WRB
reference soil group map of Ethiopia. This was presented to various partners and data-holding
institutions to demonstrate the power of data sharing. This created awareness and enabled us to
mobilise and  collate over 20,000 legacy soil profile data. These data  were then added to the national
data repository.
The data had varying levels of completeness in terms of soil field and environmental descriptions and
laboratory analysis. These required a rigorous expert-based quality assessment and standardisation
before compiling into a harmonised format. The expanded version of the Africa Soil Profile (AfSP)
database (Leenaars et al., 2014) template was used for standardising and harmonising the data. Out of
the collated soil profile data, 14,681 georeferenced data points were extracted based on completeness
and cleanness for the purposes of modelling. The cleaned soil profile data set contained, at least, the
reference soil group (RSG) nomenclature as outlined in the WRB legend. While the original soil
profile records were set in different coordinate systems, all were projected into the adopted standard
georeferencing system, namely WGS84, decimal degrees in the QGIS (3.20.2) environment (QGIS
Development Team, 2021). To verify their position, soil profile locations were plotted using a standard
WGS84 coordinate system to verify that points are matching with the site description,
geomorphological settings, and at the very least the source project boundary outline.
The accuracy of the data depends on the quality and reliability of the survey data itself which in turn
requires expert knowledge and experience in soil description and classification (Leenaars et al.,
2020a). In this study, data cleaning, validation, reclassification, and verification were carried out by a
team of prominent national pedologists and soil surveyors, including those involved in the generation
of some of the soil profile data themselves (Figure 2).

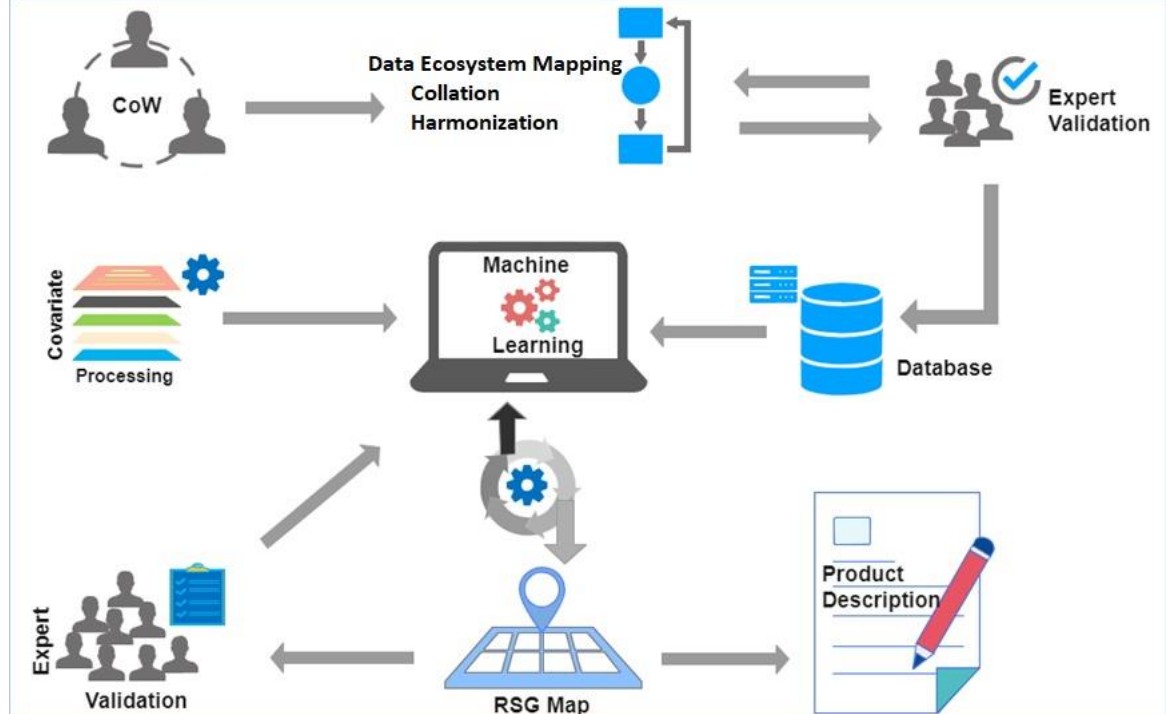

**Figure 2.** Schematic presentation of data acquisition and workflow.

In addition, the Ministry of Agriculture (MoA) soil survey and mapping experts and other volunteers have validated the legacy soil profile observations. This led to the reclassification of the soil types as deemed necessary. Such validation and reclassification involved re-examining the geomorphological setup of the soil profile locations using Google Earth as well as reviewing the site and soil descriptions and the corresponding laboratory data, and reviewing the proposed soil type. The harmonised data sets in the database were used as input soil profile data for modelling and mapping IUSS WRB reference soil groups.

## 2.3 Preparation and selection of environmental covariates

### 2.3.1 Covariates acquisition and preparation

In order to develop spatially continuous soil class/type maps, data on environmental covariates that represent directly or indirectly the soil-forming factors have to be integrated with soil profile data (Hengl and MacMillan, 2019). Environmental covariates are spatially explicit proxies of soil-forming factors based on the soil-environment relationship (McBratney et al., 2003, Shi et al., 2018). Acquisition and preparation of covariates is a crucial step in digital soil mapping using machine

learning algorithms (McBratney et al., 2003; Miller et al., 2021). In this study, 68 potential candidate
environmental variables representing soil-forming factors (climate, organisms, relief, parent material,
and time) were derived from diverse remote sensing products and thematic maps (Hengl and
MacMillan, 2019; McBratney et al., 2003).
Relief and topography-related covariates were derived from 90-meter Shuttle Radar Topography
Mission (SRTM) digital elevation model (DEM) (Vågen, 2010). Climate-related variables including
long-term mean, minimum, maximum, and standard deviation temperature, and precipitation data for
the period between 1983 and 2016 (Dinku et al., 2014) were acquired from Enhancing National
Climate Services (ENACTS-NMA) initiatives with 4 km resolutions (Dinku et al., 2014). Moderate
Resolution Imaging Spectroradiometer (MODIS) imagery raw bands and derived indices (Vågen,
2010), were downloaded from USGS EarthExplorer (https://earthexplorer.usgs.gov/) to represent
vegetation-related factors. National geological (Tefera et al., 1996), and land use and land cover
(WLRC-AAU, 2018) thematic maps of Ethiopia were gathered to represent parent material and
organisms, respectively.
Downscaling (disaggregating) or upscaling (aggregating) of rasters were also performed to match the
target resolution. A 250 m spatial resolution was chosen to accommodate both the spatial resolution
of the major covariates inputs and make it applicable for large-scale analysis. All layers were masked
for buildings and water bodies by the national boundary of Ethiopia and a stacked layer was created
using raster package (R Core Team, 2020) to extract covariate values at the locations of soil profiles.
One-hot encoding technique using dummyVars function available in Caret package (Kuhn, 2008) was
used to pre-process and convert categorical covariates into a binary vector. Each element of the binary
vector represents the presence or absence of that category. One-hot encoding is beneficial because it
allows machine-learning algorithms to interpret categorical variables as numerical features. The
covariate pre-processing, visual inspection for inconsistencies, and resampling to a target grid of 250
m were conducted in QGIS [3.20.2] (QGIS Development Team, 2021), SAGA GIS [7.8.2] (Conrad et
al., 2015) and R [version 4.05] (R Core Team, 2020) software packages. All input data were projected
to a common Lambert azimuthal equal-area projection with the latitude of origin 8.65 and centre of
meridian 39.64 which is the centre point for Ethiopia. This projection was selected since it is effective
in minimising area distortions over land. Each covariate was adjusted to have an identical spatial
resolution, extent and projection using two resampling methods. Continuous covariates were
resampled using the bilinear spline method, whereas categorical covariates were resampled using the
nearest neighbour method.

## 2.3.2 Covariates' selection

Selecting an optimal set of covariates for effectively represent the soil–environment relationship is a
key step in Digital Soil Mapping (DSM) since improper selection of covariates will affect the quality
of model outputs (Shi et al., 2018; Huang et al. 2020). In this study, near-zero variance assessment
was conducted using nearZeroVar function available in R *caret* package (Kuhn, 2008) to identify and
remove environmental variables that have little or no variance. In addition, preliminary Random Forest
model training was performed to assess and identify covariates having high variable importance. After
expert judgement, a total of 27 environmental variables (24 continuous and 3 categorical) were
selected for modelling and predicting Reference Soil Groups.

## 2.4 Modelling and mapping soil types/reference soil groups

### 2.4.1 Model tuning and quantitative evaluation
In digital soil mapping, machine-learning techniques have been extensively used to determine the
relationship between soil types and environmental variables (McBratney et al., 2003). Many machine-
learning models were developed in the past decades for digital soil mapping to spatially predict soil
classes based on existing soil data and soil-forming environmental covariates (Heung et al., 2016).
Random Forest (RF), a tree-based ensemble method, is one of the most promising machine learning
techniques available for digital soil mapping (Breiman, 2001; Heung et al., 2016), which has gained
popularity due to its high overall accuracy and has been widely used in predictive soil mapping
(Brungard, 2015; Hengl et al., 2018). Examples of the main strengths of the RF model are its ability
to handle numerical and categorical data without any assumption of the probability distribution; and
its robustness against nonlinearity and overfitting (Breiman, 2001; Svetnik et al., 2003). While
building the RF model, data was split into training (80 %) and testing (20 %) components using
random sampling for training the model and evaluating its performance, respectively (Kuhn, 2008).
Hyper-parameter optimization and repeated cross-validation on the training dataset were performed
for optimal model application using the ranger method of Caret package. The three tuning parameters
for ranger method are mtry, splitrule, and .min.node.size. Generally this function is used to tune the
parameters in modelling in an automated fashion, as this will automatically check all the possible
tuning parameters and return the optimised parameters on which the model gives the best accuracy.
Model tuning was performed with a repeated 10-fold cross-validation procedure applying multiple
combinations of hyper-parameters for the ranger method. This is a fast implementation of RF
particularly suited for high-dimensional data (Wright and Ziegler, 2017). Then the number of
covariates used for the splits (mtry), splitting rules (splitrule) and minimum node size (min.node.size)
were optimised. The parameter ntree was adjusted to 1,000 in the model, and mtry values (10, 15, 20),
min.node.size values (5, 10, 15), and splitrule values ("variance'', "extratrees'', and "maxstat") were
fed  for the optimization procedure. The accuracy of the testing dataset was related to the model
performance for the new dataset, indicating the capacity of the model to predict at the unsampled
location. A confusion matrix was also used to calculate a cross-tabulation of observed and predicted
classes with associated statistics i.e., producer's accuracy and user's accuracy.
**2.4.2 Software and computational framework**
In this study, various open-source software packages that provide a comprehensive set of tools and
diverse capabilities were used for data preparation, analysis and visualisation. Data pre-processing and
preparation were performed using QGIS (QGIS Development Team, 2021) and SAGA GIS (Conrad
et al., 2015). For statistical analysis and machine learning modelling, R (R Core Team, 2020) and
relevant libraries were installed on a Windows server 2016 standard with 250 GB of working memory
to handle the challenges associated with large-scale data processing and analysis.
**2.4.3 Expert evaluation of spatial patterns of the beta-version soil map**
Visual inspection of the DSM output over the terrain was used to identify abnormalities and assess
how effectively it depicts landscape components (Rossiter et al., 2022). For this, we employed an
expert-based qualitative assessment of the model output. This technique was used to complement
model-based accuracy assessment and confirm agreement or indicate areas of concern. This was
implemented by a panel of senior soil specialists/pedologists checking the map based on purposely
selected district level geographic windows across Ethiopia, representing different agro-ecological
zones known to have diverse soil occurrences, and familiar to the panel of experts. Accordingly, an
expert validation workshop was conducted using the first version of the reference soil groups (RSGs)
map. About 45 multi-disciplinary scientists including soil surveyors, pedologists, geologists, and
geomorphologists were drawn from national and international research, development, and higher
learning institutions to review the draft RSG map in plenary. This was followed by breakout sessions
where groups of experts evaluated the map based on their experience and knowledge of soil-landscape
relations of the country and examined geographic windows.
Most importantly, disagreements regarding RSGs occurrence and patterns of the modelling outputs
across topo-sequences and contrasting soil-forming  factor sequences were identified and discussed.
Further, inferences on parts of the DSM framework that require improvement were recommended.
After finalising the evaluation at the group's level assessment, each group presented the results in the
plenary followed by a discussion to get feedback from other participants. Following the plenary
discussions, the participants created a group of six senior pedologists to work on the recommendations
including  changing the quality mask layer, validation of the additional data obtained during the event,
and assessment of re-modelling outputs.
After the second model was re-run, the group of senior pedologists together with  geospatial experts
re-evaluated the output using the selected districts based on the feedback from the first review, which
was mainly on areas where there were  "minor" and "major" concerns. Consequently, some
improvements were made e.g., in the areas where Vertisols, Fluvisols, and Leptosols were
overestimated. Further, underestimated RSGs (Alisols, Solonetz, Planosols, Acrisols, Lixisols,
Phaeozems, and Gleysols) showed a slight increase in area coverage and pattern improvements.
However, the total area of Leptosols and Cambisols increased from the first run due to the partial
exclusion of the mask layer used in the first round of modelling.  The mask layer used in the first run
was criticised for quality issues as it excluded significant soil areas and due to its weakness in
capturing non-soil areas such as rock outcrops, salt flats, swamps and sand dunes. Nevertheless, the
spatial patterns of these soils occurring across previously considered "non-soil areas'' were examined
by the panel of experts. In parallel, geospatial and soil experts checked the raster map of the RSGs in
the GIS environment to ensure areas with 'no concern' before re-running the model are kept the same
or changes are accepted by the panel of experts. The map from the second run is presented in this
paper as EthioSoilGrids version 1.0 product.


## 3 Results and Discussion

### 3.1 Soil profile datasets

Using the IUSS WRB, 2015, the preliminary identified 14,742 georeferenced legacy soil profiles were classified/reclassified into twenty-three reference soil groups (RSGs). Nearly 90% of the soil profile points represented Vertisols, followed by Luvisols, Cambisols, Leptosols, Fluvisols, and Nitisols, which were found to be the dominant soil types in Ethiopia (Figure 3). The remaining 10% represented the Regosols, Alisols, Andosols, Arenosols, Calcisols, Solonetz, Lixisols, Phaeozems, Solonchaks, Acrisols, Planosols, Gleysols, Umbrisols, Ferralsols, Gypsisols, Plinthosols, and Stagnosols.

According to this study, about 72% of the IUSS WRB (2015) RSGs were confirmed to occur in Ethiopia. This reconfirms the characterization of Ethiopia as a land of soil diversity having endowed with a diverse range of soil types (Elias, 2016; Mishra et al., 2004). One of the limitations with legacy soil data in categorical mapping is the imbalanced soil samples, in that all classes were not equally represented (Wadoux et al., 2020). For this study, soil profiles with less than 30 observations were objectively excluded from the model after examining the accuracy and spatial distribution of each reference soil group. Five reference soil groups (Umbrisols, Ferralsols, Gypsisols, Plinthosols, and Stagnosols) were excluded from the model and the EthioSoilGrids version 1.0 map.

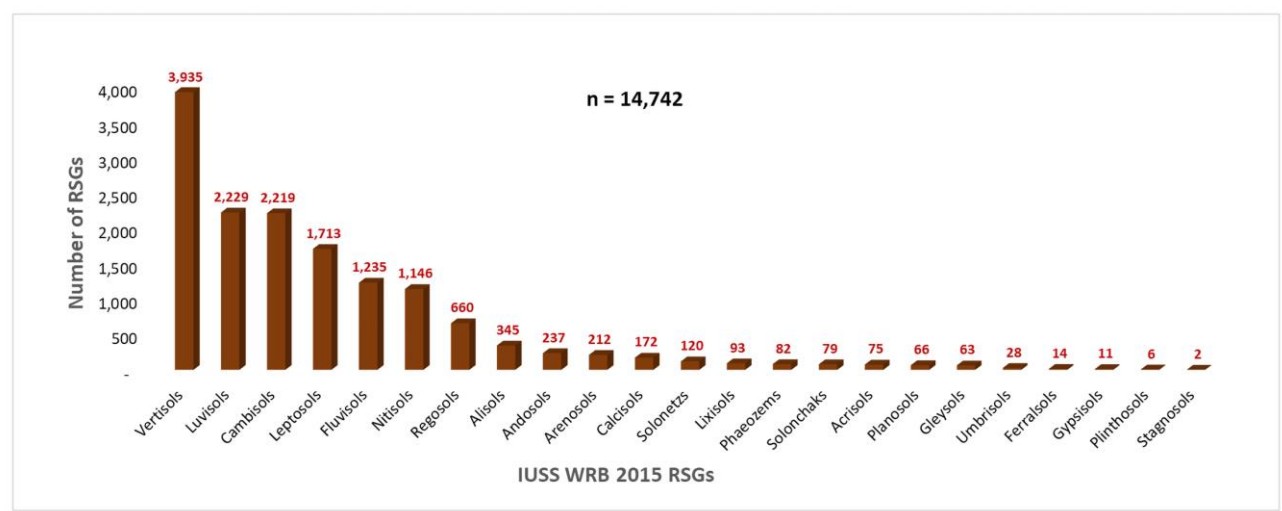

**Figure 3.** Number of soil profile points per WRB reference soil groups.

After excluding the built-up) and water surface areas the average soil profile density was 13.1 per 1,000 km² (Figure 4), but the actual density varied  across the different parts of the country. The variation tends to follow river basins, sub-basins, and agricultural land-use types-based studies from which most of the legacy data  were pulled.. For instance, in 30 intervention districts of the Capacity Building for Scaling up of Evidence-Based Best Practices in Agricultural Production in Ethiopia (CASCAPE) project, the average profile density was about 87 profiles per 1,000 $km^2$ for a total area of about 26,830 km² (Leenars et al., 2020a). Similarly, semi-detailed soil mapping missions in 15 districts conducted through the Bilateral Ethiopia-Netherlands Effort for Food, Income and Trade (BENEFIT)-REALISE project generated about 217 observations per 1,000 km² (Leenars et al., 2020b).

A soil type and depth map compilation and updating mission at a 1:250,000 scale by the Water Land Resource Centre (WLRC) of Addis Ababa University collated and used about 3,949 legacy soil profiles for the entire country (Ali et al., 2020), which is about 3.5 profiles per 1,000 $km^2$.  Although the distribution is not even and the eastern lowlands are sparsely represented, the number of data used in this study is 8.5 times higher than the 1,712 legacy soil profiles data currently existing in the Africa soil profile database (Batjas et al., 2020; Leenaars et al., 2014).

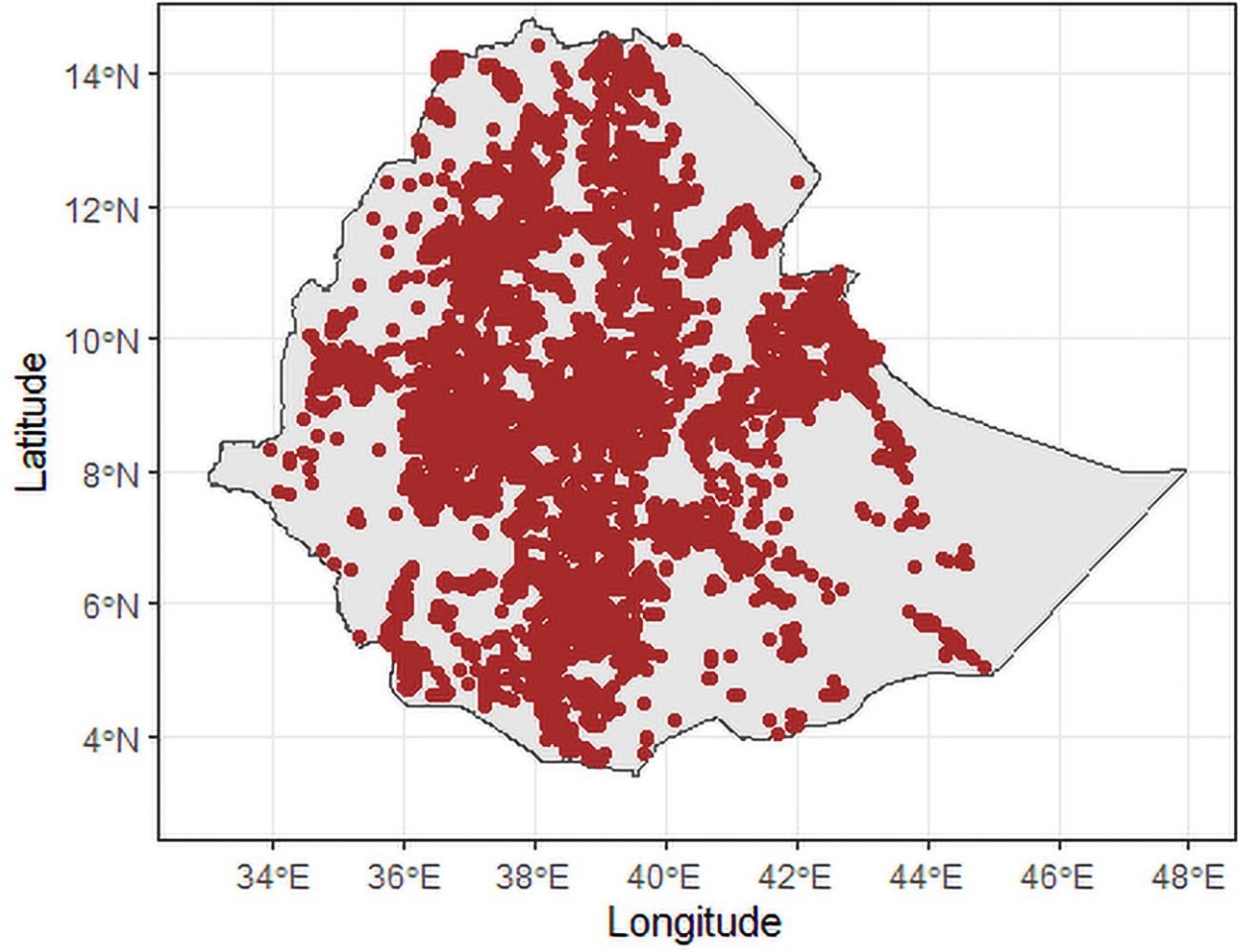

**Figure 4.** Spatial distribution of collated legacy soil profile data.

The soil profiles distribution across the 32 agro-ecological zones (AEZ) of Ethiopia revealed that all, except two–tepid per-humid mid-highland (0.13% landmass) and very cold sub-humid sub-afro alpine to afro-alpine (0.03% landmass)–were represented by soil profile observations. Furthermore, about 95% of the profile observations represented 91% of the AEZs aerial coverage (Appendix A). The distribution of legacy soil profiles varied across AEZs. In general, the top-ranked lowland AEZs with roughly 56% area coverage were represented by 23% of the total profile observations, whereas top-ranked highland AEZs with 20% area coverage received 47% of profile observations. For instance, warm desert, warm moist, hot arid, and warm sub-moist lowlands with area coverage of around 20%, 15%, 11%, and 10%, were represented roughly by 3%, 11%, 2%, and 7% of the total profiles, respectively. Tepid moist mid highlands (8% area coverage), tepid sub-humid mid highlands (7% area

coverage), and tepid sub-moist mid highlands (5% area coverage) each were represented by 20%, 15%, and 12% of the profiles, respectively.

## 3.2 Modelling and Mapping

### 3.2.1 Variable importance

The reference soil group spatial pattern is primarily influenced by long-term average surface reflectance, flow-based DEM indices, and precipitation. Figure 5 shows variables of importance for determining RSGs spatial prediction. The top-ranked variables were (i) long-term MODIS Near-Infrared (NIR) reflectance; (ii) multiresolution index of valley bottom flatness, (iii) long-term mean day-land surface temperature; (iv) long-term mean soil moisture; (v) standard deviation of long-term precipitation; (vi) long-term mean precipitation; and (vii) topographic wetness index.

MODIS long-term mean spectral signatures showed high relative importance. According to Hengl et al. (2017), accounting for seasonal vegetation fluctuation and inter-annual variations in surface reflectance, long-term temporal signatures of the soil surface, derived as monthly averages from long-term MODIS imagery, were more effective. Furthermore, Hengl and MacMillan (2019) explained that long-term average seasonal signatures of surface reflectance provide a better indication of soil characteristics than only a single snapshot of surface reflectance.

The Multi-Resolution Valley Bottom Flatness Index, a DEM-derived topography index, is the second top-ranked covariate driving soil variability across Ethiopia. This hydrological/soil removal and accumulation/deposition index is used to distinguish valley floor and ridgetop landscape positions (Soil Science Division Staff, 2017) highly responsible for multiple soil-forming processes to operate over a particular landscape, resulting in a wide range of soil development. The influence of topography on spatial soil variation is manifested in every landscape of Ethiopia (Belay, 1997; Mesfin, 1998; Nyssen et al., 2019; Zewdie, 2013).

Long-term daily mean land surface temperature, mean soil moisture, rainfall standard deviation and mean annual rainfall were among the top-ranked covariates for predicting reference soil groups' spatial variation across the country. In Ethiopia, different soil genesis studies revealed that climate has a significant influence on soil development and properties and is, therefore, responsible for having widely varying soils in the country (Abayneh, 2006, 2005; Fikru, 1988, 1980; Zewdie, 2013).

359

Among the most important covariates for predicting reference soil groups in the Ethiopian highlands, are monthly average soil moisture for January (ranked 3$^{rd}$), long-term average soil moisture (ranked 4$^{th}$), and monthly average soil moisture for August (ranked 5$^{th}$) (Leenars et al., 2020a). In the current study, soil moisture was among the ten top ranked covariates in modelling and explaining long-distance soil type variability across the country.

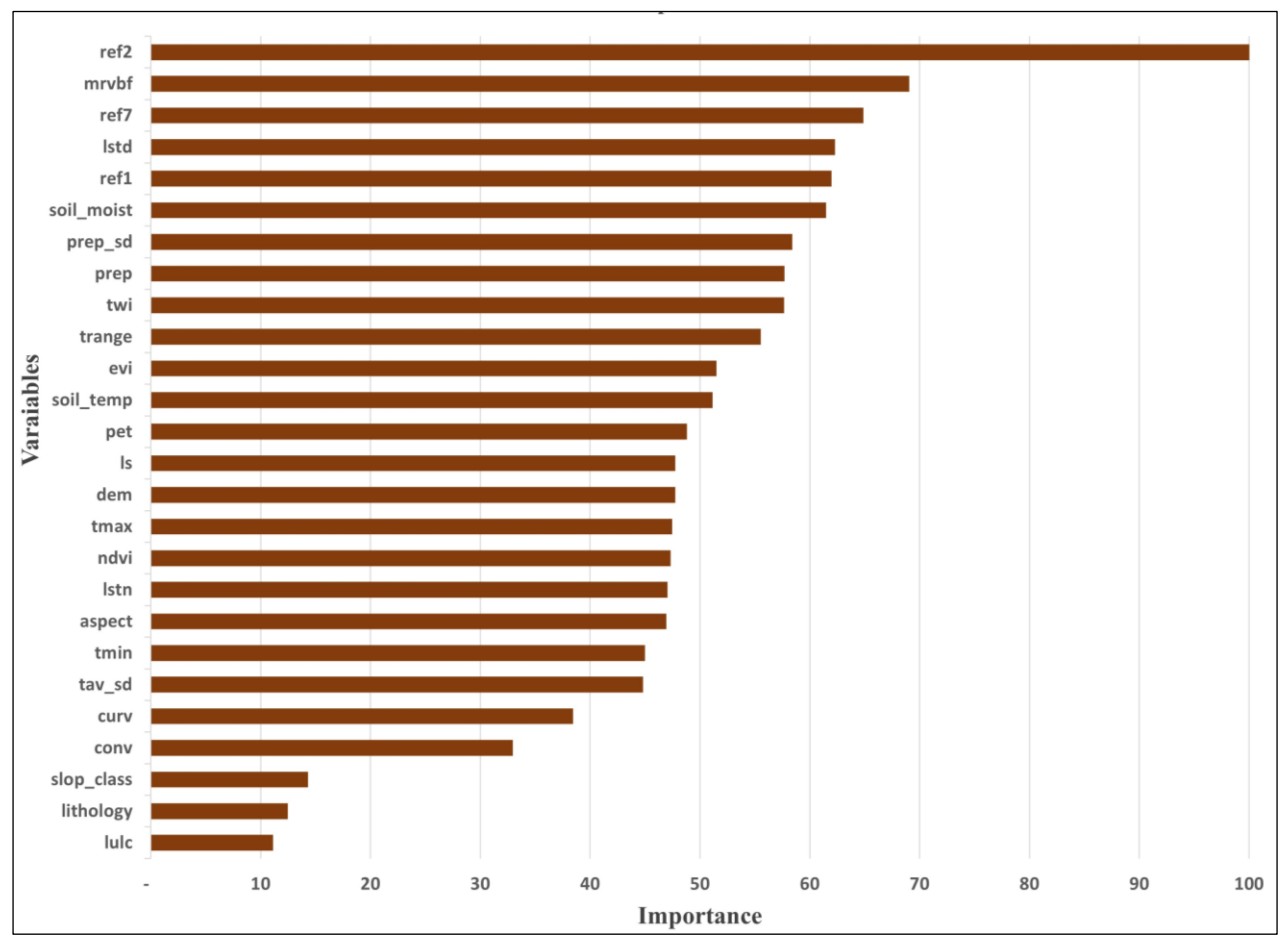

**Figure 5.** Random forest covariate relative importance for modelling RSGs.
Note: prep=Precipitation; prep_sd=The standard deviation of precipitation; tmax=Maximum Temperature; tmin=Minimum Temperature; trange=Temperature range; tav_sd=Standard deviation of average temperature; pet=Potential evapotranspiration; lstd=Land surface temperature- Day; lstn=Land surface temperature-Night; soil_moist=Soil Moisture ; soil_temp=Soil temperature; DEM =Digital elevation model (Elevation); twi =Topographic wetness Index; aspect=Topographic Aspect; curv=Topographic Curvature; conv=Topographic convergence index; ls=Slope Length and Steepness

factor (ls_factor); morph=Terrain Morphometry; mrvbf=Multiresolution index of valley bottom flatness; slope=Slope class (%); ndvi=Normalised Difference Vegetation Index (NDVI); evi=Enhanced Vegetation Index (EVI); lulc=Land use/ landcover; lithology=Geology; ref1=Red band ;ref2=Near-Infrared; ref7=Mid-Infrared.

In this study, lithology showed a relatively low influence on soil variability may be due to the use of a coarse-scale and less detailed lithology map, which may not sufficiently capture the spatial variability of the parent materials.

### 3.2.2 Model performance

The parameter optimization process resulted in mtry = 20, split rule= extra trees and minimum node size= 5. The overall accuracy of the model was 56.24% which ranged between 54.43% and 58.1% with a 95% confidence interval. The kappa values based on the internal cross-validation and testing dataset showed that the overall model performance produced using 10–fold cross-validation with the repeated fitting was 48%. Considering similar area-based digital soil class mapping efforts, the overall accuracy was in line with the accuracies that were typically reported for soil class maps developed with random forest model (Leenaars et al., 2020a) and statistical methods (Heung et al., 2016; Holmes et al., 2015). Table 1 shows the confusion matrix at validation/testing points i.e., 20 % of the observation. Further, the matrix indicates the producer's accuracy (class representation of observed versus predicted) and user's accuracy were not similar for all RSGs. The map purity is in the order of Lixisols, Calcisols, Alisols, Phaeozems, Vertisols, Andosols, Solonchaks, Fluvisols, Arenosols, Leptosols, Luvisols, Nitisols, and Cambisols. However, Vertisols, Calcisols, and Andosols are the observed classes that are best represented by the map followed by Fluvisols, Alisols, Nitisols, Leptosols, Luvisols and Cambisols.

Global Soil Grids at 250 m resolution used machine learning algorithms to map the global WRB reference soil groups with map purity and weighted kappa of 28% and 42%, respectively (Hengl et al., 2017). The Soil Grids 250 m WRB soil groups/classes prediction output-spatial soil patterns were not evaluated based on expert knowledge while in this study we did an extensive back and forth qualitative assessment by a panel of pedologists. The quantitative accuracy in the present study (about 56%) coupled with an expert-based qualitative evaluation of the predicted maps indicated the development and achievement of a substantially enhanced national product for users of spatial soil resource

information. This finding is a step forward and acceptable considering that Soil Grids are not expected
to be as accurate as locally produced maps and models that use much more local point data and finer
local variables (Mulder et al., 2016). Further, the data and findings in this study can help improve the
soil maps of Africa as it partially addresses the concern by Hengl et al. (2017) who recognised that
WRB RSGs modelling in the global Soil Grids 250 m is critically uncertain for parts of Africa. This
is mainly attributed to limited access to more local point data by regional and global modelling
initiatives, unlike the present study which accessed a large number of legacy soil profile datasets.


**Table 1.** Confusion matrix of random forest RSG prediction (at validation/testing observations).

| | | | | | | | Reference | | | | | | | | | | | | | |
|---|---|---|---|---|---|---|---|---|---|---|---|---|---|---|---|---|---|---|---|---|
| Prediction | Acrisols | Alisols | Andosols | Arenosols | Calcisols | Cambisols | Fluvisols | Gleysols | Leptosols | Lixisols | Luvisols | Nitisols | Phaeozems | Planosols | Regosols | Solonchaks | Solonetzs | Vertisols | User Accuracy | Total |
| Acrisols | 1 | 0 | 0 | 0 | 0 | 0 | 0 | 0 | 0 | 0 | 1 | 0 | 0 | 0 | 0 | 0 | 1 | 0 | 0.33 | 3 |
| Alisols | 0 | 40 | 0 | 0 | 0 | 0 | 1 | 1 | 0 | 0 | 9 | 4 | 0 | 0 | 2 | 0 | 0 | 2 | 0.68 | 59 |
| Andosols | 0 | 0 | 28 | 1 | 1 | 3 | 5 | 0 | 2 | 0 | 2 | 0 | 0 | 0 | 0 | 0 | 1 | 1 | 0.64 | 44 |
| Arenosols | 0 | 0 | 0 | 11 | 0 | 2 | 1 | 0 | 0 | 0 | 5 | 0 | 0 | 0 | 0 | 0 | 0 | 1 | 0.55 | 20 |
| Calcisols | 0 | 0 | 0 | 0 | 21 | 0 | 1 | 0 | 0 | 0 | 2 | 0 | 0 | 0 | 0 | 0 | 0 | 5 | 0.72 | 29 |
| Cambisols | 2 | 3 | 6 | 9 | 1 | 197 | 28 | 2 | 35 | 2 | 47 | 16 | 5 | 1 | 16 | 3 | 3 | 28 | 0.49 | 404 |
| Fluvisols | 1 | 0 | 3 | 5 | 1 | 34 | 144 | 0 | 9 | 0 | 15 | 7 | 0 | 0 | 1 | 5 | 5 | 17 | 0.58 | 247 |
| Gleysols | 0 | 0 | 0 | 0 | 0 | 0 | 1 | 2 | 0 | 0 | 1 | 0 | 0 | 1 | 0 | 0 | 0 | 0 | 0.40 | 5 |
| Leptosols | 0 | 1 | 4 | 3 | 3 | 47 | 11 | 0 | 176 | 0 | 27 | 7 | 1 | 0 | 32 | 0 | 0 | 24 | 0.52 | 336 |
| Lixisols | 0 | 0 | 0 | 0 | 0 | 0 | 0 | 0 | 0 | 1 | 0 | 0 | 0 | 0 | 0 | 0 | 0 | 0 | 1.00 | 1 |
| Luvisols | 2 | 16 | 3 | 8 | 0 | 34 | 13 | 2 | 33 | 3 | 216 | 30 | 3 | 0 | 25 | 1 | 0 | 41 | 0.50 | 430 |
| Nitisols | 6 | 8 | 0 | 0 | 1 | 23 | 8 | 3 | 18 | 8 | 29 | 132 | 0 | 1 | 8 | 0 | 1 | 21 | 0.49 | 267 |
| Phaeozems | 0 | 0 | 0 | 0 | 0 | 0 | 0 | 0 | 0 | 0 | 1 | 0 | 2 | 0 | 0 | 0 | 0 | 0 | 0.67 | 3 |
| Planosols | 0 | 0 | 0 | 0 | 0 | 0 | 0 | 0 | 0 | 0 | 1 | 1 | 0 | 5 | 1 | 0 | 0 | 1 | 0.55 | 9 |
| Regosols | 0 | 0 | 0 | 0 | 0 | 7 | 1 | 0 | 7 | 1 | 8 | 1 | 0 | 0 | 22 | 0 | 0 | 5 | 0.42 | 52 |
| Solonchaks | 0 | 0 | 0 | 0 | 0 | 0 | 0 | 0 | 1 | 0 | 0 | 0 | 0 | 0 | 0 | 3 | 1 | 0 | 0.60 | 5 |
| Solonetzs | 0 | 0 | 0 | 0 | 1 | 4 | 1 | 0 | 0 | 0 | 0 | 0 | 0 | 0 | 0 | 1 | 6 | 0 | 0.46 | 13 |
| Vertisols | 3 | 1 | 3 | 5 | 5 | 92 | 32 | 2 | 61 | 3 | 81 | 31 | 5 | 5 | 25 | 2 | 6 | 641 | 0.64 | 1,003 |
| Producer Accuracy | 0.07 | 0.58 | 0.60 | 0.26 | 0.62 | 0.44 | 0.58 | 0.17 | 0.51 | 0.06 | 0.49 | 0.58 | 0.13 | 0.38 | 0.17 | 0.20 | 0.25 | 0.81 | 0.56 | - |
| Total | 15 | 69 | 47 | 42 | 34 | 443 | 247 | 12 | 342 | 18 | 445 | 229 | 16 | 13 | 132 | 15 | 24 | 787 | - | 2,930 |

### 3.2.3 Modelling and Mapping: EthioSoilGrids Version 1.0

The study identified eighteen reference soil groups in Ethiopia, mapped at 250 m resolution (Figure 6). The model prediction showed that seven soil reference groups including Cambisols, Leptosols, Vertisols, Fluvisols, Nitisols, Luvisols, and Calcisols covered nearly 98% of the total land area of the country (Figure 7). Five soil reference groups (Solonchaks, Arenosols, Regosols, Andosols, and

Alisols) were estimated to cover about 2% of the land area, while trace coverages of Solonetz, Planosols, Acrisols, Lixisols, Phaeozems, and Gleysols were also found in some pocket areas.

In terms of spatial distribution, Nitisols and Luvisols dominated the northwestern and southwestern highlands while the southeastern lowlands were dominantly covered by Cambisols, Calcisols, and Fluvisols with some Solonchaks. The Vertisols extensively cover the north and south-western lowlands along with the Ethio-Sudan border areas and central highland plateaus. The probability of occurrence of each RSG was mapped (Appendix C) in each modelling spatial window (i.e., the cell size of 250-meter X 250 m). The dominant RSGs were aggregated based on the most probable RSGs in each spatial modelling window. There was high correspondence between the top seven ranked prediction probabilities and observed soil types as confirmed visually by overlaying observed classes and prediction probabilities.

The overall occurrence and the relative position of each of the RSG along the topo-sequence and its association with other RSGs agree with previous works (Abayneh, 2006; Ali et al., 2010; Abdenna et al., 2018; Asmamaw and Mohammed, 2012; Belay, 2000, 1998, 1997, 1996; Driessen et al., 2001; Elias, 2016; FAO 1984a; Fikre, 2003; Mitku, 1987; Mohammed and Belay, 2008; Mohammed and Solomon, 2012; Mulugeta et al., 2021; Nyssen et al., 2019; Sheleme, 2017; Shimeles et al., 2007; Tolossa, 2015; Zewdie, 2013). However, in some cases, the RSGs' position along the topo-sequence and association with other RSGs require further investigation. The observed disparities might be attributed to the positional accuracy of legacy point observations, modelling approach, and most importantly the level of detail and scale/resolution of the environmental variables used in this study. We used the currently available coarse resolution national geological map and hence soil parent material might be inadequately represented in the model, which probably resulted in irregular RSGs sequences. For instance, the main driving factors to establish and explain soil-landscape variability in May-Leiba catchment of northern Ethiopia were geology (soil parent material) and different mass movements (Van de Wauw et al., 2008). These factors led to Cambisols– Vertisols catenas on basalt and Regosols–Cambisols–Vertisols catenas on limestone formations. Similar studies identified parent material strongly determines the soil type (e.g. Vertisol, Luvisol, Cambisol) (Nyssen et al., 2019). In general, in areas where there is complex soil diversity and distribution of soils, one of the most important parameters is to identify parent material including effective techniques to capture

and delineate mass movement bodies, and human-induced soil erosion and deposition areas (Leenars
et al., 2020a;  Nyssen et al., 2019; Van de Wauw et al., 2008).

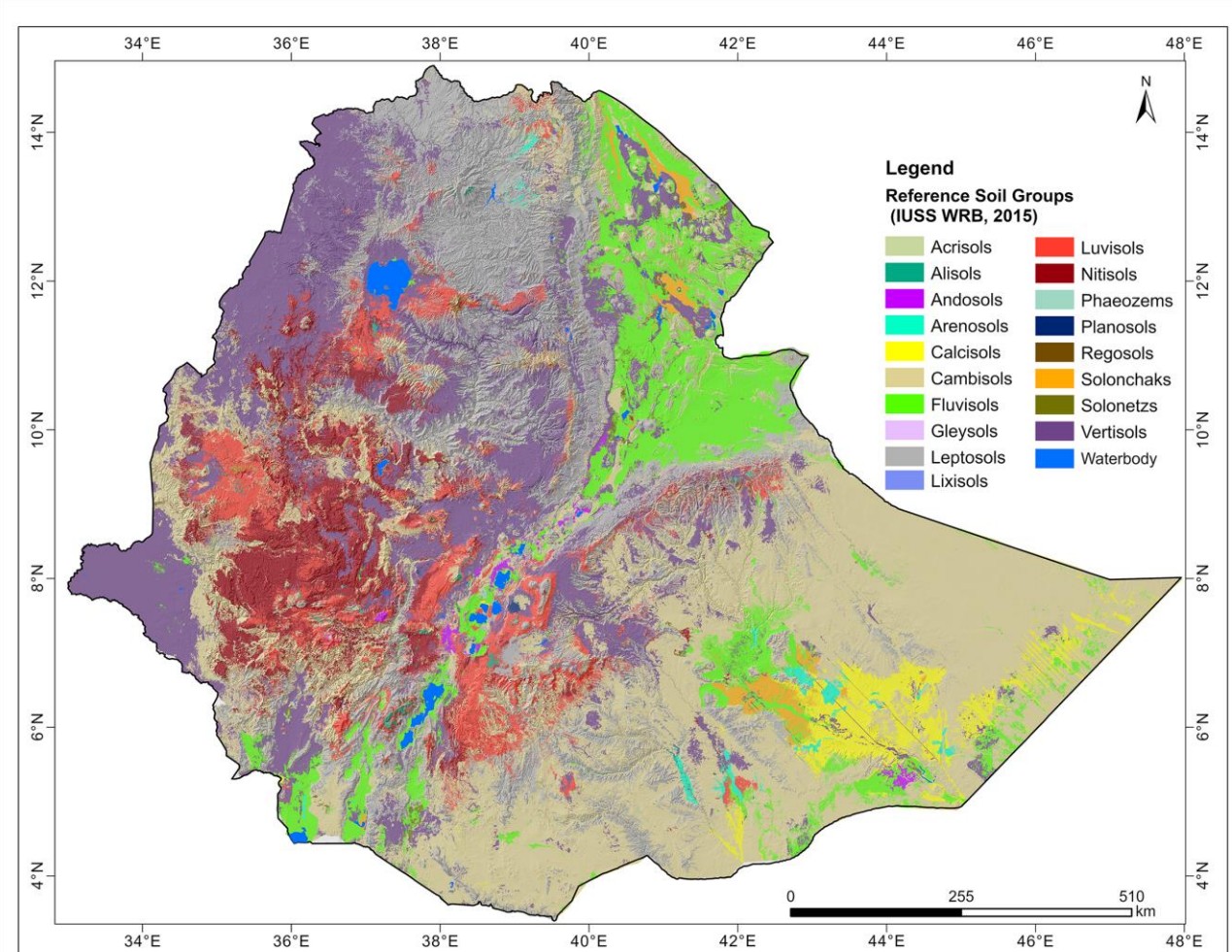

**Figure 6.** Major reference soil groups of Ethiopia (EthioSoilGrid V1.0).
Considering the third position of Cambisols in the order of frequency occurrence of RSGs per point
observations (following Vertisols and Luvisols), these soils seem to be over-represented on the map
(ranked 1st) apparently at the expense of Vertisols and Luvisols, and to some extent in places of
Leptosols and other RSGs. This might be attributed to the fact that Cambisols create a geographical
continuation with Vertisols and/or Luvisols at the lower slopes and Leptosols/ Regosols at the higher
slopes, suggesting the presence of some bordering soil qualities in respective transitional zones (Ali et
al., 2010; Asmamaw and Mohammed, 2012; Sheleme, 2017; Zewdie, 2013).
The proportion of area mapped as Cambisols (34 %) revealed new insights compared with the
information from the most cited spatial soil maps:  Cambisols ranked 2nd (21 %), 2nd (16 %), 4th (9 %),
and 4th (8 %) as reported by Berhanu (1980), FAO (1984b), FAO (1998), and Soil Grids- Hengl et al
(2017), respectively. This might be due to: (i) the number and distribution of profile observations,
which is more extensive than the previous ones, (ii) the type and level of details of covariates
considered; (iii) variations and rearrangements in the keys for classification of the RSGs among soil
classification versions used in previous studies and misclassification/confusion of Vertisols with
Vertic Cambisols, as legacy soil profile data coming from diverse sources.

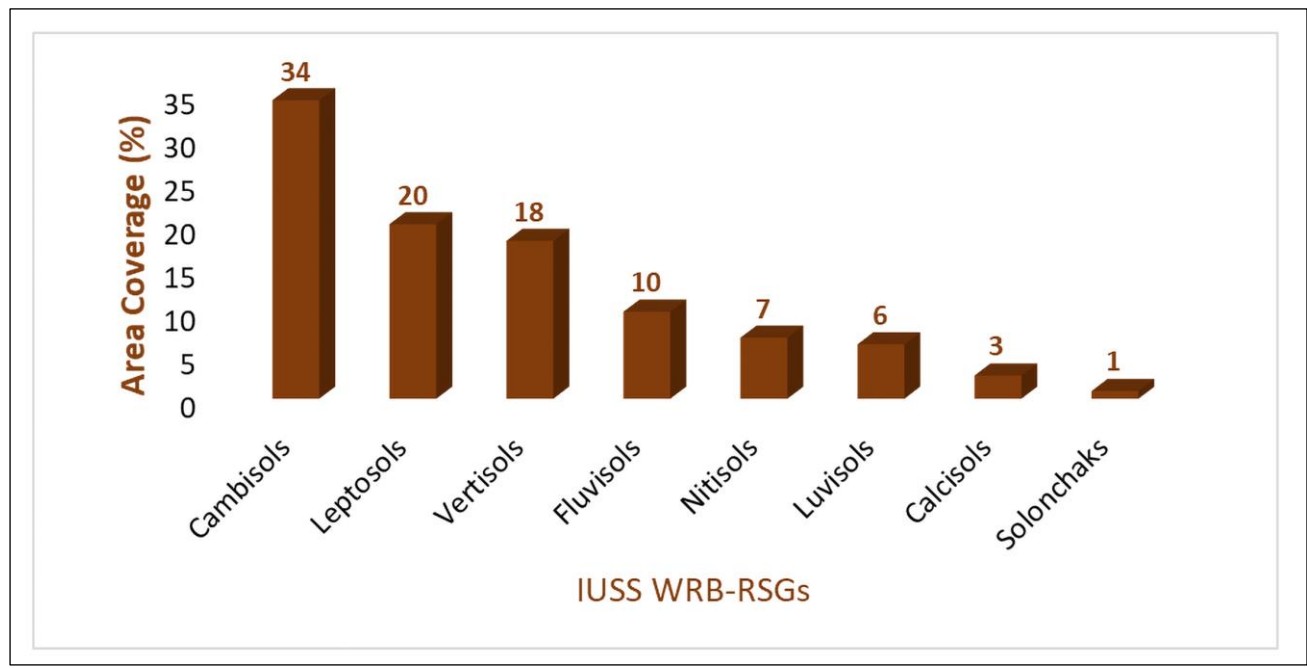

**Figure 7.** The area coverage (in %) for the major WRB RSGs (Note: the remaining 10 RSGs-
Arenosols (0.44 %), Regosols (0.35 %), Andosols (0.31 %), Alisols (0.16 %), Solonetzs (0.04 %),
Planosols (0.04 %), Acrisols (0.02 %), Lixisols (0.02 %), Phaeozems (0.02 %), and Gleysols (0.01 %)
were not plotted because of their relatively small area coverage).

**3.3 Expert validation of the soil map**

Expert knowledge of soil-landscape relations and soil distribution remains important to evaluate the predictive soil mapping results and assess if predicted spatial patterns make sense from a pedological viewpoint (Hengl et al., 2017; Poggio et al., 2020; Rossiter et al., 2022). An important step in qualitative model evaluation is, therefore, expert assessment whereby professionals with broad experience in soil survey and mapping can evaluate and improve the quality of the soil resource map. This can highlight areas of agreement or concern across the landscape (Rossiter et al., 2022). The expert validation workshop provided useful insights and tangible improvements to the development of the map. While the plenary discussion provided an overview of the approaches followed in developing the map, the group discussions helped to have an in-depth review of the selected polygons of the map assigned to them. Participants were split into five groups (with 8-10 members each) and have chosen up to 60 polygons representing areas with which at least one of the group members has sufficient information, including data sources. Overall, the groups have checked a total of 126 polygons (Figure 8) which were fairly distributed across the country.

487

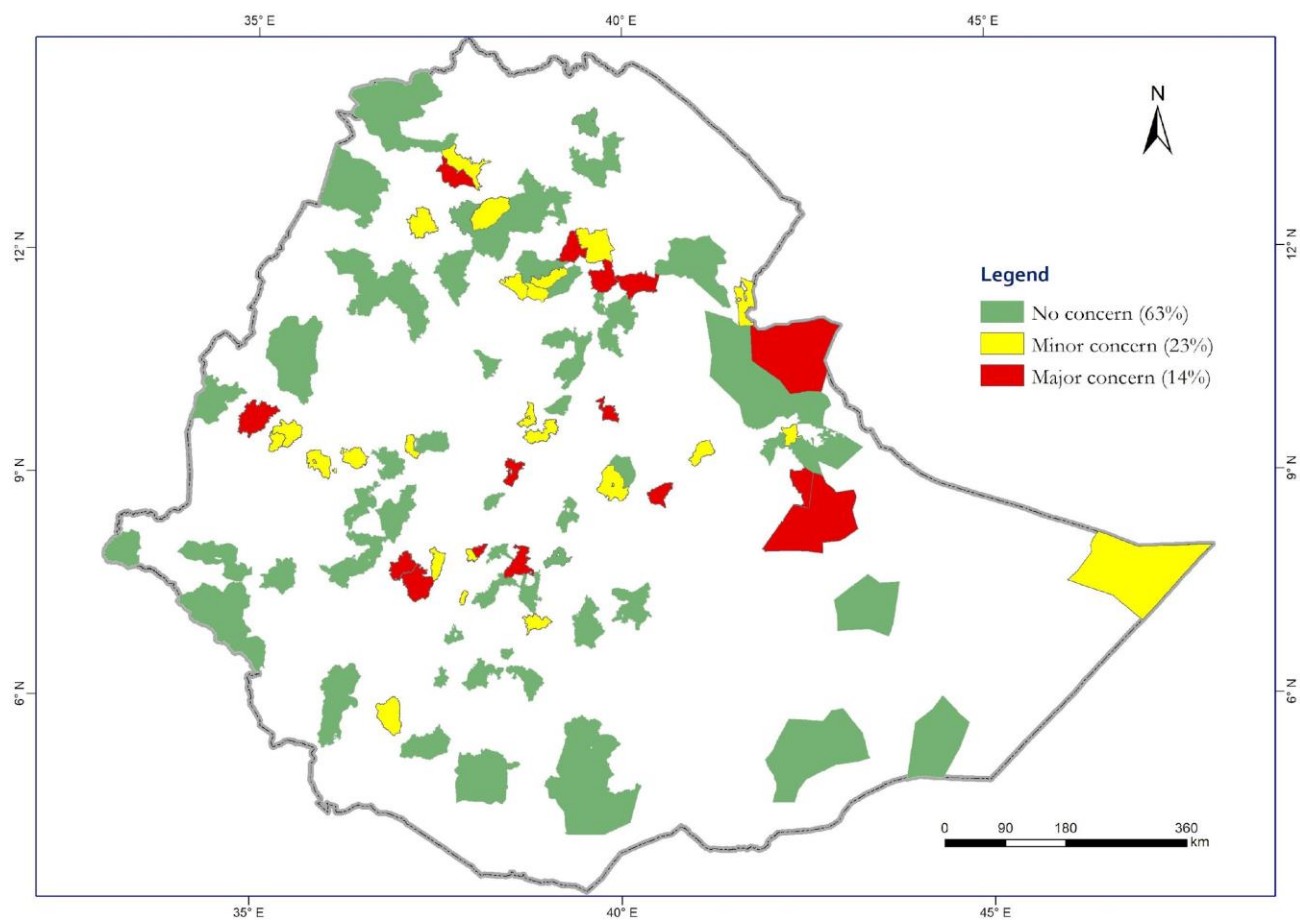

**Figure 8.** The spatial distribution of districts validated by stakeholders and feedback categories
according to the level of concerns raised.

The group members displayed the polygons one by one in a GIS environment and discussed the
predicted dominant and associated soil reference soil groups and labelled them in one of three
confirmation categories: 1. confirmed with '*no concern'*, 2. confirmed with "*minor concern''*, and 3.
confirmed with '*major concern'*. Confirmation with '*no concern'* was made when all members of a
group agreed on both the types, relative coverage and patterns of the predicted soils within the polygon.
Confirmation with '*minor concern*' was made when all or some of the team members agreed on the
predicted soil types within the polygons but did not agree on the order of abundance or the probability
occurrence of one or two soils including observed spatial patterns, while confirmation with '*major*
*concern*' was made when all members of the team did not agree on the predicted soil type, or when
the presence of another soil type, other than the predicted ones is noted.

All three groups have rated the accuracy of the map at 60 +%; of the 126 polygons, they have expressed
no concern for 63 %, minor concern for 23 % and a major concern for 14 % of the polygons.
Furthermore, differences in the prevalence of RSGs and patterns of the modeling outputs across
different soil forming factor sequences, as well as inferences about which areas of the DSM framework
still need work, were identified  and elaborated by the expert input, and presented in the subsequent
sections.
**3.4 Evaluation of results, limitations and future direction**
Up-to-date soil resource spatial information is critically missing at a required scale and extent in
Ethiopia. As a result, resource management strategies miss their targets. Furthermore, the absence of
such data at a required resolution and extent, forced decision support tool developers to pick and use
the data they can access and afford. As a result, model outputs appear more site-specific or
representation becomes homogenous over the very heterogeneous landscapes that exist in reality. On
the other hand, in large areas and complex landscapes such as Ethiopia, it is very difficult to address
the demand for reasonably  accurate and detailed soil-type  maps using a conventional approach due
to the costs involved, and resources  and time it requires. For instance, given the vastness of the country
and heterogeneous  landscapes, a new conventional soil survey mission requires at least 170,000
profile point observations to map the entire terrestrial land mass of Ethiopia at a scale of 1: 250,000
with at least 1 observations per square centimetre. Moreover, the soil profile data requirement
definitely could have been much higher as we increase the scale of mapping and density of
observations. In the present study, machine-learning techniques combined with expert input were
implemented to produce a countrywide soil resource map of Ethiopia at reasonably higher accuracy,
less time and cost than that of conventional methods. In addition, rescue, compilations and
standardization of about 14,681 geo-referenced legacy soil profiles that can be included in the National
Soil Information System (NSIS) of Ethiopia and the World Soil Information Centre will support future
national, regional and global DSM efforts. The approach used demonstrates the power of data and
analytics to map the soil resources of Ethiopia and the output is an exemplary use case for similar
digital content development efforts in Ethiopia and beyond.
Moreover, in this study the quality monitoring processes and methods were followed to filter dubious
soil profiles, and soil classification and harmonization protocols. Then after, the study followed a
robust modelling framework and generated new insights into the relative area coverage of WRB RSGs
of Ethiopia. In addition, the study provided coherent and up-to-date digital quantitative gridded spatial
soil resource information to support the successful implementation of various digital agricultural
solutions and decision support tools (DSTs).
The spatially explicit limitation of the present study is revealed by expert-based qualitative evaluation
of spatial patterns across objectively selected geographic windows and prominent contrasting
landscapes of Ethiopia. This qualitative assessment indicated areas of concern in terms of how well
EthioSoilGrids version 1.0 represents soil geography across a mosaic of the country's landscapes. For
instance, in the north-eastern lowlands of Ethiopia, mainly along the "Denakil" depression, Fluvisols,
Cambisols and Vertisols were found on the map in areas where normally other soil types were expected
to occur. In this area, the expected prediction and area coverage of Leptosols has been probably
overshadowed by Fluvisols and Cambisols. Similarly, in some parts of western Ethiopia landscapes,
the prediction of Vertisols overshadows other RSGs which resulted in area coverage underestimation
of Fluvisols (along the "Akobo", "Gilo", and "Baro" rivers and their tributaries) and Alisols. Likewise,
in the central parts of northwestern Ethiopia, the prediction of Nitisols has been overshadowed by
Vertisols and Luvisols resulting in probable underestimation of the Nitisols area coverage.
The relatively low  model performance and some classification errors in some of the  examined
geographic windows (e.g. the Denakil depression , along Akobo, Baro, and Gilo rivers and the Somali
region) is , probably due  to the paucity of samples from those areas (Figure 4), the inadequacy of the
dataset by RSGs, and over-representation of the dataset by some RSGs such as Vertisols, Luvisols,
and Cambisols. Balanced datasets are ideal to allow a decision tree algorithms to produce better
classification but for datasets with uneven class size, the generated classification model might be
biased  towards the majority class (Hounkpatin et al., 2018; Wadoux et al., 2020). In addition,
uncertainty around quality of included covariates, not considered covariates in the modelling process
including management, use of validation methods that do not sufficiently control the effect of clustered
samples, and small sample size for some RSGs could have possibly biased modelling results in some
geographic areas.
To improve the modelling performance, future studies could explore (1) adding data for under-
represented geographic areas, land uses and covariate spaces, (2) opportunities to include other
covariates (parent material and management) that could capture the variability of the country
heterogeneous landscapes, (3) dimension reduction of covariates (4) use of remedial measures for
imbalances in sample sizes, (5) comparing different cross-validation methods, (6) use of an ensemble
modelling approach and/or robust modelling technique that accommodates neighbourhood size and
connectivity analyses, (7) use of better resolution/quality mask layer to segregate non-soil areas (rock
outcrops, salt flats, sand dunes and water bodies) from mapping areas, and (8) implementation of
quantitative and qualitative comparison of national, regional, and global legacy soil maps/soil grids
with new DSM products in terms of how well DSM products represent soil geography. In addition ,
future digital soil mapping strategies in Ethiopia may require  to consider new soil sampling missions
in under-represented areas, adopt standard soil sampling, description guidelines and soil classification
systems including soil physico-chemical and mineralogical analysis, and combine local soil
nomenclature/classification systems with RSGs and develop a map of RSGs with qualifiers. At the
moment the under-sampled and under-represented areas are the Somali region, the Denakil and the
western and northwestern border areas of Ethiopia (Figure 4). Regardless of these limitations and to
the best of our knowledge the EthioSoilGrids v1.0 product provides the most complete soil information
available for Ethiopia.
**4 Conclusions**
Coherent and up-to-date country-wide digital soil information is essential to support digital
agricultural transformation efforts. This study involved collation, cleaning, harmonization, and
validation of the legacy soil profile data sets, involving soil scientists with different backgrounds
individually and in groups. To develop the 250 m digital soil resource map, a machine learning
modelling approach and expert validation were applied to the harmonised soil database and
environmental covariates affecting soil-forming processes. Accordingly, about 20,000 soil profile data
have been collated, out of which, about 14,681 were used for the modelling and mapping of eighteen
RSGs out of the identified twenty-three RSGs. Although unevenly distributed, the legacy soil profile
data used in the modelling covered most of the agro-ecologies of the country.
Among the mapped 18 RSGs, the highest number of observed (3,935) profiles represent Vertisols,
followed by Luvisols, Cambisols and Leptosols, while Gleysols were represented with the lowest
number (63) of profiles. The modelling revealed that MODIS long-term  reflectance, multiresolution
index of valley bottom flatness, land surface temperature, soil moisture, long-term mean annual
rainfall, and wetness index of the landscape are  the most important covariates for predicting reference
soil groups in Ethiopia.
Our ten-fold spatial cross-validation result showed an overall accuracy of about 56 % with varying
accuracy levels among RSGs. The modelling result revealed that seven major soil reference groups
including Cambisols (34 %), Leptosols (20 %), Vertisols (18 %), Fluvisols (10 %) Nitisols (7 %),
Luvisols (6 %) and Calcisols (3 %) covered nearly 98 % of the total land area of the country, while
minor coverage of other reference soil groups (Solonchaks, Arenosols, Regosols, Andosols, Alisols,
Solonetzs, Planosols, Acrisols, Lixisols, Phaeozems, and Gleysols) were also detected in some areas.
Compared to the existing soil resource map, the coverage of the first three major soil groups has
substantially increased which is related to the increased availability of soil profile data covering larger
areas of the country, implying that these soils were previously underestimated. Cambisols and
Vertisols which together represent nearly half of the total land area are relatively young with inherent
fertility, implying the high agricultural potential for the country. However, given their limitations,
these and the other soil types require the implementation of suitable land, water, and crop management
techniques to sustainably exploit their potential.
The EthioSoilGrids version 1.0 product from this first countrywide RSGs modelling effort requires
complementary activities. These include modelling and mapping that should go beyond RSGs and
need to include $2^{nd}$ level classifications including principal and supplementary qualifiers. Furthermore,
soil atlas of Ethiopia with details of the soil physicochemical properties needs to be prepared together
with the map, for which the authors and/or others responsible need to prioritize in their future research
endeavours.



 **Appendix A: Legacy soil profile data distribution**

 **Table A1.** Distribution of legacy soil profile data by agroecology zones.

| Major agroecological zones | AEZ area coverage (%)* | Profiles observation (%)** |
|---|---|---|
| Warm arid lowland plains | 19.76 | 3.40 |
| Warm moist lowlands | 15.12 | 10.74 |
| Hot arid lowland plains | 10.79 | 2.44 |
| Warm sub-moist lowlands | 9.63 | 6.94 |
| Tepid moist mid highlands | 8.05 | 20.21 |
| Warm sub-humid lowlands | 7.11 | 5.69 |
| Tepid sub-humid mid highlands | 6.63 | 15.26 |
| Tepid sub-moist mid highlands | 5.17 | 12.39 |
| Warm semi-arid lowlands | 2.75 | 3.23 |
| Tepid humid mid highlands | 2.65 | 2.48 |
| Warm humid lowlands | 2.29 | 0.45 |
| Cool moist mid highlands | 1.74 | 4.15 |
| Hot sub-humid lowlands | 1.67 | 0.07 |
| Cool sub-moist mid highlands | 1.16 | 3.00 |
| Cool  humid mid highlands | 0.82 | 1.01 |
| Warm per-humid lowlands | 0.68 | 0.01 |

| Major agroecological zones | AEZ area coverage (%)* | Profiles observation (%)** |
|---|---|---|
| Hot moist lowlands | 0.59 | 3.56 |
| Hot sub-moist lowlands | 0.56 | 0.03 |
| Cool sub-humid mid highlands | 0.52 | 1.38 |
| Tepid arid mid highlands | 0.43 | 0.39 |
| Hot semi-arid lowlands | 0.40 | 2.05 |
| Tepid semi-arid mid highlands | 0.19 | 0.67 |
| Cold moist sub-afro-alpine to afro-alpine | 0.07 | 0.16 |
| Cold sub-moist mid highlands | 0.07 | 0.04 |
| Cold sub-humid sub-afro-alpine to afro-alpine | 0.06 | 0.03 |
| Cold humid sub-afro-alpine to afro-alpine | 0.06 | 0.01 |
| Very cold humid sub-afro-alpine | 0.04 | 0.02 |
| Very cold sub-moist mid highlands | 0.02 | 0.02 |
| Very cold moist sub-afro-alpine to afro-alpine | 0.01V | 0.03 |
| Hot per-humid lowlands | 0.01 | 0.15 |
| Tepid perhumid mid highland | 0.13 | 0 |
| Very cold sub-humid sub-afro alpine to afro-alpine | 0.03 | 0 |

Note: *= total area of Ethiopia 1.14mln km$^2$ ; **=total number of profiles 14,681

 **Appendix B: Environmental covariates**

 **Table B1.** List, description, spatial and temporal extent, and source of covariates used in modelling
 the reference soil groups.

| Categories | Covariates | Descriptions | Spatial resolution | Temporal resolution | Source |
|---|---|---|---|---|---|
| Climate | prep | Precipitation | 4 km | 1981 - 2016 | ENACTS (Dinku et al.,2014) |
|  | prep_sd | The standard deviation of precipitation | 4 km | 1981 - 2016 | Derived from ENACTS (Dinku et al.,2014) |
|  | tmax | Maximum Temperature | 4 km | 1983 - 2016 | ENACTS (Dinku et al.,2014) |
|  | tmin | Minimum Temperature | 4 km | 1983 - 2016 | ENACTS (Dinku et al.,2014) |
|  | trange | Temperature range | 4 km | 1983 - 2016 | ENACTS (Dinku et al.,2014) |
|  | tav_sd | Standard deviation of average temperature | 4 km | 1983 - 2016 | Derived from ENACTS (Dinku et al.,2014) |
|  | pet | Potential evapotranspiration | 4 km | 1981 - 2016 | Derived from ENACTS (Dinku et al.,2014) using Modified Penman method |
|  | lstd | Land surface temperature- Day (Aqua MODIS- MYD11A2 , time series monthly average) | 1000 m | 2002-2018 | AfSIS [a] |
|  | lstn | Land surface temperature-Night (Aqua MODIS- MYD11A2 , time series monthly average) | 1000 m | 2002-2018 | AfSIS |
|  | soil_moist | Soil Moisture (Derived from one-dimensional soil water balance) | 4 km | 1981 - 2016 | Ethiopian Digital AgroClimate Advisory Platform (EDACaP) |
|  | soil_temp | Soil temperature | 30 km | 1979 - 2019 | ERA 5-Reanalysis ECMWF data [b] |
| Topography | DEM | Digital elevation model (Elevation) | 90 m | - | SRTM- DEM (Vågen, 2010) |

| Categories | Covariates | Descriptions | Spatial resolution | Temporal resolution | Source |
|---|---|---|---|---|---|
| | twi | Topographic wetness Index | 90 m | - | SAGA GIS-based SRTM-DEM derivative |
| | aspect | Topographic Aspect | 90 m | - | SAGA GIS-based SRTM-DEM derivative |
| | curv | Topographic Curvature | 90 m | - | SAGA GIS-based SRTM-DEM derivative |
| | conv | Topographic convergence index | 90 m | - | SAGA GIS-based SRTM-DEM derivative |
| | ls | Slope Length and Steepness factor (ls_factor) | 90 m | - | SAGA GIS-based SRTM-DEM derivative |
| | morph | Terrain Morphometry | 90 m | - | SAGA GIS-based SRTM-DEM derivative |
| | mrvbf | Multiresolution index of valley bottom flatness | 90 m | - | SAGA GIS-based SRTM-DEM derivative |
| | slope | Slope class (%) | 90 m | - | SAGA GIS-based SRTM-DEM derivative |
| Vegetation | ndvi | Normalised Difference Vegetation Index (NDVI) (MODIS- MODIS MOD13Q1, time series monthly average) | 250 m | 2000-2021 | AfSIS [a] |
| | evi | Enhanced Vegetation Index (EVI) (MODIS- MODIS MOD13Q1, time series monthly average) | 250 m | 2000-2021 | AfSIS |
| | lulc | Land use/ landcover | 30 m | 2010 | Water and Land Resource Centre-Addis Ababa University (WLRC-AAU, 2010) |
| parent material | lithology | Geology/parent material | 1:2,000,000 | 1996 | The Ethiopian Geological Survey (Tefera et al.,1996) |
| MODIS spectral refelectance | ref1 | Red band (MODIS- MODIS MOD13Q1, time series monthly average) | 250 m | 2000 – 2018 | AfSIS [a] |
| | ref2 | Near-Infrared (MODIS- MODIS MOD13Q1, time series monthly average) | 250 m | 2000 – 2018 | AfSIS |

| Categories | Covariates | Descriptions | Spatial resolution | Temporal resolution | Source |
|---|---|---|---|---|---|
|  | ref7 | Mid-Infrared (MODIS- MODIS MOD13Q1, time series monthly average) | 250 m | 2000 – 2018 | AfSIS |


**Appendix C:** Probability of occurrence of reference soil groups

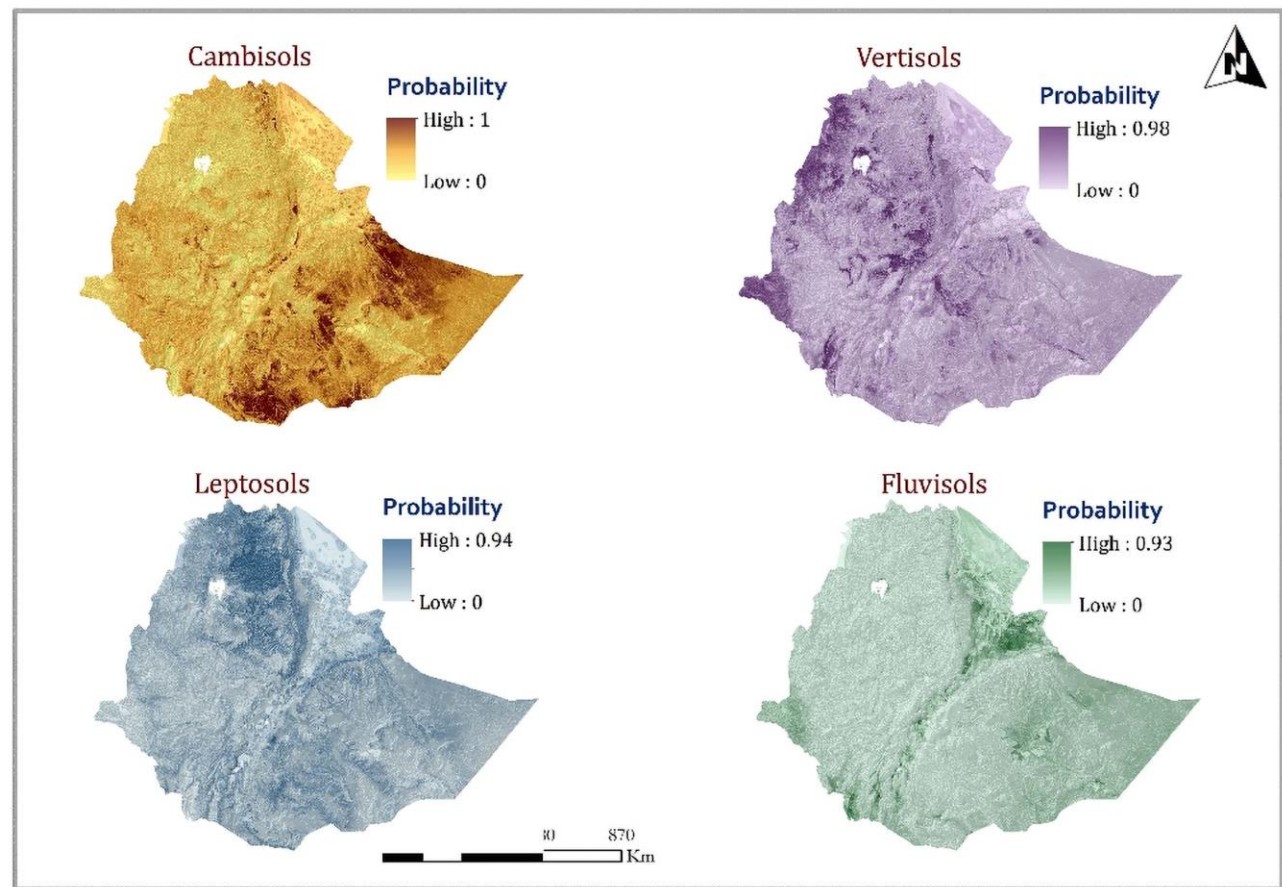

**Figure C1.** Occurrence probability maps of Cambisols, Leptosols, Vertisols, and Fluvisols.


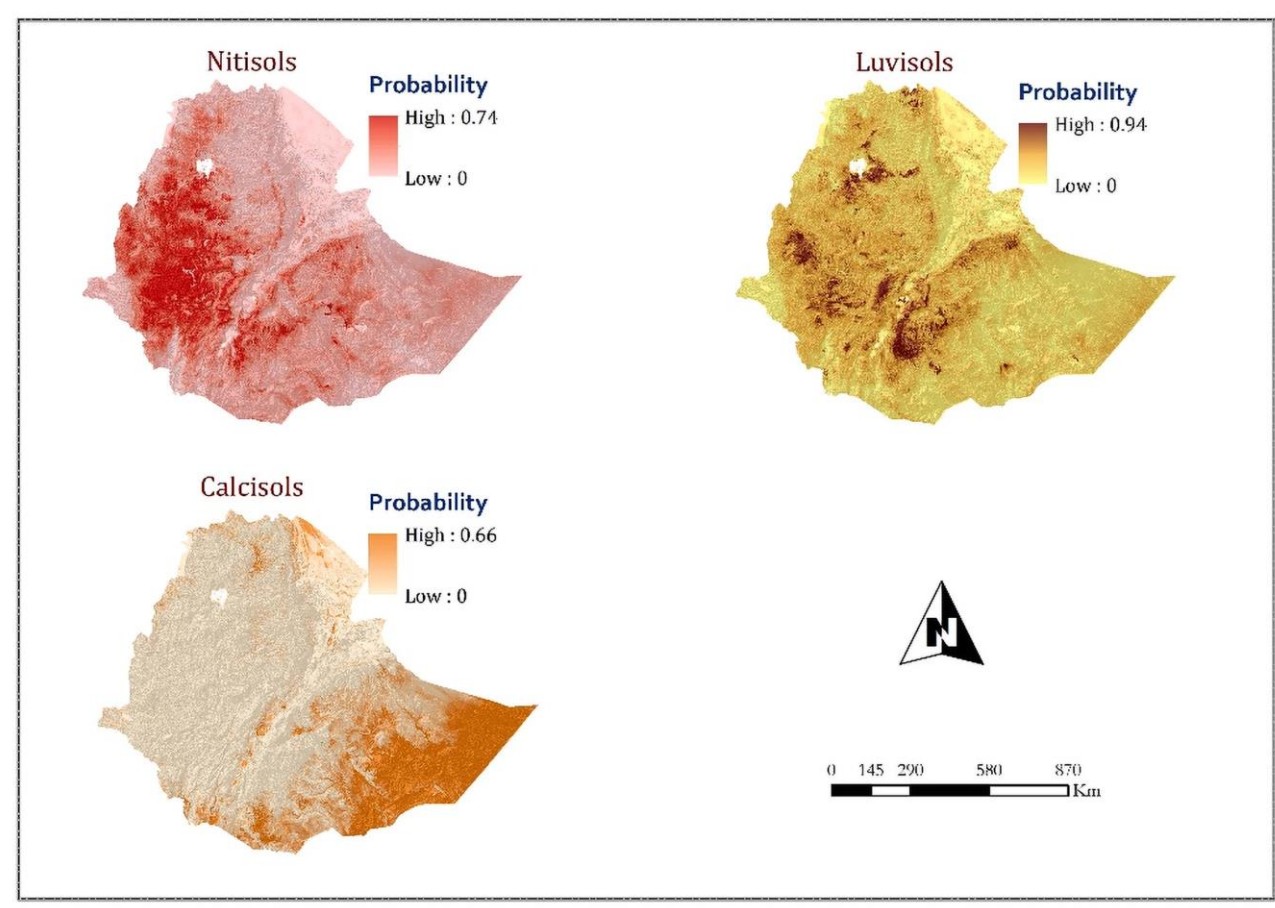

**Figure C2.** Occurrence probability maps of Nitisols, Luvisols, and Calcisols.


**Data availability**. Full data will be available upon request based on the CoW guideline (CoW, 2020; https://ethioagridata.com/) and the MoA "Soil and Agronomy Data Management, Use and Sharing" directive No. 974/2023 Ethiopia (https://nsis.moa.gov.et/).

**Author contributions.** AA, TE, KG, WA, and LT conceived and designed the study, perform the analysis, and wrote the first draft, with substantial input and feedback from all authors. EM, TM, NH, AY, AM, TA, FW, AL, NT, AA, SG, YA,  and BA,  contributed to input data preparation, data encoding, and harmonization. Legacy data validation and review of subsequent versions of the paper were performed by MH, WH, AA, DT, GB, MG, SB, MA, AR, YGS, ST, DA, YW, DB, EZ, SC, and EE.

**Competing interests.** The authors declare that they have no conflict of interest.

**Acknowledgments.** We sincerely appreciate the coalition of the willing (CoW) members who are instrumental in providing, collating, cleaning, standardizing and harmonizing the legacy soil profile data used in generating the soil resource map of Ethiopia at 250 m resolution. The CoW team also deserves credit for inspiring many to share data and develop an integrated national database related to agronomy and soil profile data. The leadership of the Natural Resource Development Sector and Soil Resource Information and Mapping Directorate of the Ministry of Agriculture (MoA) have played a crucial role. These includes assigning experts from the Ministry and other organizations who worked on collating, encoding, harmonizing, processing the soil survey legacy data, and modelling and prediction of Reference Soil Groups using robust machine learning algorithms and high performance computing servers  are the foundation for the soil resource map. Various institutions, as well as the late and present soil surveyors and pedologists, deserve special recognition for their contributions to the generation and sharing of soil profile data. We owe a debt of gratitude to ISRIC and the bilateral Ethio-The Netherlands projects (cascape and BENEFIT-REALISE) funded by the Directorate-General for International Cooperation (DGIS) of the Netherlands Ministry of Foreign Affairs through the Netherlands Embassy in Ethiopia, which have been crucial in providing capacity building to the MoA, and national soil and geospatial experts. Much thanks are due to Professor Eyasu Elias, Arie van Kekem,  Dr Tewodros Tefera, Dr Mulugeta Diro, Johan Leenaars, Bas Kempen, Stephan Mantel, and Maria Ruiperez Gonzalez who have been organizing and providing training on soil classifications and digital soil mapping to the MoA, and national soil and geospatial experts, over the Ethio-The Netherlands bilateral projects' period. The senior pedologists and soil surveyors who provided invaluable support to check and harmonize thousands of soil profiles and laboratory results are sincerely appreciated. They worked very hard with positive energy for which we are very grateful. In addition, the same group of experts and additional ones who supported the validation of the preliminary soil resource map deserve credit for their commitment to contributing their experiences. We thank Dr Degefe Tebebe , Dr Sileshi Gudeta, and Neil Munro for supporting in the extraction of climate covariates , providing critical technical support , and comments that helped improve the paper. Our sincere appreciation also goes to the continued and persistent support of GIZ-Ethiopia mainly through the project- Supporting Soil Health Interventions in Ethiopia (SSHI), which supported and facilitated the activities of the CoW. The Alliance of Bioversity and CIAT is highly acknowledged for coordinating CoW and its efforts and supporting the implementation of activities that are of high national importance. We would also like to sincerely thank the Excellence in Agronomy (EiA) CGIAR

Initiative, which has brought huge contributions to this project in terms of funding and building skill
of the various teams. The Water, Land and Ecosystems (WLE) and Climate Change, Agriculture and
Food Security (CCAFS) programs of the CGIAR also provided support in various forms. Recently,
our work is benefiting from the Accelerating Impacts of CGIAR Climate Research in Africa
(AICCRA) project supported by the World Bank in terms of data, analytics, and resources to support
data linkage and integration.
**Financial support**. This work is financially supported by the Deutsche Gesellschaft für Internationale
Zusammenarbeit (GIZ) through the project "Supporting Soil Health Interventions in Ethiopia," funded
by the Bill and Melinda Gates Foundation. This work was supported, in whole or in part, by the Bill
& Melinda Gates Foundation [INV-005460]. Under the grant conditions of the Foundation, a Creative
Commons Attribution 4.0 Generic License has already been assigned to the Author Accepted
Manuscript version that might arise from this submission.

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
