# Peer review of "Reference Soil Groups Map of Ethiopia Based on Legacy Data and Machine"

_EGUsphere, 2022_

## Community Comment (CC3)

**Suggestions for improvement**

While I appreciate the effort made by the authors in responding to my comments, I still believe they have not addressed the main problems, i.e., (1) mismatch between their map (i.e., EthioSoil Grid map) and the Soil Atlas of Africa, and (2) relatively low accuracies of classification. The EthioSoil Grid map is also inconsistent with the SOTER Map of Ethiop (Figure 3.3) produced by Elias (2016)[1]. In my opinion it is better to direct efforts towards identifying the reasons for these discrepancies and fine-tuning the analysis accordingly to produce a map that achieves greater accuracy. The overall accuracy of 56.2% reported in the paper does not inspire confidence. The Kappa coefficient of 48% is even less comforting; it tells us that the classification is not better than can be found by chance alone (random). If Kappa was higher (e.g., >60% or more, we could confidently say that the classification was significantly better than a random association. I extracted the producer and user accuracy from Table 1 of Ahenafi et al and calculated the omission and commission errors for each reference soil groups (RSGs) (see table below). Evidently, the errors of classification are large (orange) or unacceptable (red) for most RSGs. The reason is partly a sampling problem. Note in the table below and the following graph that as the sample size for RSGs decreased, the omission and commission errors increased. Even with large sample sizes some RSGs (e.g., Luvisols and Cambisols have been classified with >50% error.

| Reference soil group | Sample size (N) | Accuracy | | Error | | |
| --- | --- | --- | --- | --- | --- | --- |
| | | Producer | User | Omission | Commission | Comment |
| Vertisols | 3935 | 0.81 | 0.64 | 19 | 36 | Acceptable error (AE) |
| Cambisols | 2219 | 0.44 | 0.49 | 56 | 51 | Unacceptable error (UE) |
| Luvisols | 2229 | 0.49 | 0.50 | 51 | 50 | UE |
| Leptosols | 1713 | 0.51 | 0.52 | 49 | 48 | Large (LE) |
| Fluvisols | 1235 | 0.58 | 0.58 | 42 | 42 | LE |
| Nitisols | 1146 | 0.58 | 0.49 | 42 | 51 | LE |
| Regosols | 660 | 0.17 | 0.42 | 83 | 58 | UE |
| Alisols | 345 | 0.58 | 0.68 | 42 | 32 | LE |
| Andosols | 237 | 0.60 | 0.64 | 40 | 36 | LE |
| Arenosols | 212 | 0.26 | 0.55 | 74 | 45 | UE |
| Calcisols | 172 | 0.62 | 0.72 | 38 | 28 | AE |
| Solentz | 120 | 0.25 | 0.46 | 75 | 54 | UE |
| Lixisols | 93 | 0.06 | 1.00 | 94 | 0 | UE due to very small producer N = 1 |
| Phaeozeme | 82 | 0.13 | 0.67 | 87 | 33 | UE due to very small user N |
| Soleschaks | 79 | 0.20 | 0.60 | 80 | 40 | UE due to very small user N |
| Acrisols | 75 | 0.07 | 0.33 | 93 | 67 | UE due to very small user N |
| Planosols | 66 | 0.38 | 0.11 | 62 | 89 | UE due to very small user N |
| Gleysols | 53 | 0.17 | 0.40 | 83 | 60 | UE due to very small user N |
| Umbrisols | 28 | | | | | |
| Ferralsols | 14 | | | | | Conditions in Ethiopia do not favour |
| Gypsisols | 11 | | | | | |
| Plinthosols | 6 | | | | | Conditions in Ethiopia do not favour |
| Stagnosols | 2 | | | | | |

Omission error = 100-producer accuracy (in %); Commission error = 100-user accuracy (in %);

Note that the discrepancy between omission and commission errors is higher when sample sizes are small as in Gleysols, Planosols, Acrisols, Solenshaks, Phaeozemes and Lixisols. Note also in the figure below that both omission and commission errors vary widely when sample sizes are small, but they tend to be small with increasing sample size. That means that for the map to be improved there is a need for soil sampling in

[1] Eyasu Elias, 2016. Soils of the Ethiopian Highlands: Geomorphology and Properties. CASCAPE Project, ALTERA, Wageningen University and Research Centre (Wageningen UR). The Netherlands. 385pp

the areas not covered in this analysis. No amount of modelling sophistication can compensate for small sample size.

[Figure]

Trends in omission and commission errors with sample size. Red lines represent smoothing splines

Even for Vertisols, which had the largest sample size, the commission error is large (36%), and the probability that the points on the map will actually be present on the ground are lower than we are made to believe. Note that the producer's accuracy for the Vertisols is 81% but the user accuracy is 64%. This means that although 81% of the reference Vertisols have been correctly identified as "Vertisols" on the map, only 64% of the areas identified as "Vertisols" in the classification were actually Vertisols. A good example of this problem is revealed by the following maps covering the Western part of Ethiopia. The EthioSoil Grid map shows an entire region covered purely by Vertisols. According to the Soil Atlas of Africa, large areas are covered by Fluvisols (along the Akobo, Gilo, Baro Rivers and their tributaries). Another discrepancy between the Soil Atlas of Africa and the EthioSoil Grid map is absence of Alisols in this region. I believe these discrepancies were caused by inadequacy of legacy data available to the authors (see Figure 5 in Ashenafi et al.). Note that Alisols are one of the most problematic soils (Al toxicity, low pH, susceptibility to drought) and their misclassification in this map can create problems for their sustainable management. A third problem with the EthioSoil Grid map is that it shows other soils (Acrisols? Cambisols?) in areas typically dominated by Nitisols. Nitisols are prone to acidification with fertilizer use, and their misidentification as other soils may create problems for fertilizer management.

[Figure]

The classification of Cambisols and Luvisols is also problematic. Note that the producer and user accuracy for Cambisols are 44% and 49%, respectively. This means that even though 44% of the reference Camisols have been correctly identified as "Cambisols", only 49% of the areas identified as "Camisols" in the classification were actually Cambisols. The omission error (56%) and commission errors (51%) are too big. Examination of the Confusion Matrix reveals that Cambisols were misclassified in large number of cases as Luvisols (47), Leptosols (35), Fluvisols (28), Vertisols (28), Nitisols (16), Regosols (16), etc. Similarly, Leptosols were misclassified as Cambisols (47), Regosols (32), Luvisols (27), Vertisols (24), Fluvisols (11), etc. That is probably why Cambisols and Fluvisols were found on the map in areas where normally other soil types were expected to occur. For example, the following maps clearly shows the mismatch between the Soil Atlas of Africa and the EthioSoil Grid map (Ashenafi et al.). I suspect the mismatch was partly caused by inadequacy of legacy data from the region mapped here (see Figure 5 in Ashenafi et al.). Evidently, EthioSoil Grid map shows Fluvisols and Vertisols? even in mountainous areas, where conditions for the genesis of Fluvisols and Vertisols do not exist in this region. The Soil Atlas of Africa, on the other hand, shows Fluvisols only around water bodies, where Fluvisols are likely to develop. It also clearly shows other RSGs including the qualifiers (divisions within RSGs), for example as Calcic Fluvisols, Eutric Fluvisols, Salic Fluvisols). That kind of detail is more useful to decision-makers than just saying Fluvisols.

[Figure]

The following maps shows another huge discrepancy between the Soil Atlas of Africa and EthioSoilGrid. Here, EthioSoilGrid shows Cambisols and Fluvisols covering a large proportion of the land although Cambisols and Fluvisols are least expected. I suspect the mismatch was partly caused by inadequacy of legacy data from this region (see data gaps in Figure 5 of Ashenafi et al.). Indeed, the sample size for Gypsisols was very small. So, I also suspect that Gypsisols were not included in the training dataset. Therefore, the algorithm misclassified the RSGs as Cambisols based on the information it has. In machine learning this is called **association bias**. As far as your machine learning model is concerned, Gypsisols do not exist.

[Figure]

**Dominant:** Calcisols, Gypsisols, Regosols
**Minor:** Vertisols, Arenosols

**Dominant:** Cambisols, Calcisols, Fluvisols
**Minor:** Vertisols

Soil Atlas of Africa

EthioSoilGrids 1.0

In summary, I believe that inadequacy of sample sizes for some RSGs, paucity of samples from some areas and the over-representation of the dataset by a few RSGs (e.g., Vertisols, Cambisols and Luvisols), has probably led to large classification errors. In addition, the covariates included/not included in the machine learning algorithm and the use of validation methods which do not sufficiently control overfitting could bias results with small sample size. Simulation studies[2] show that K-fold Cross-Validation produces strongly biases performance with small sample sizes, even with sample size of 1000. There is a strong need for closely looking at the previous analysis and redoing the map so that it can be used as a reliable decision support tool. I encourage the authors to explore (1) opportunities to include other variables not included in the present analysis, (2) dimension reduction, (3) use of other cross-validation methods, and (4) use of an ensemble approach to see whether overall accuracy could be improved and classification errors reduce for individual RSGs.

[2] Vabalas A, Gowen E, Poliakoff E, Casson AJ (2019) Machine learning algorithm validation with a limited sample size. PLoS ONE 14(11): e0224365. https://doi.org/10.1371/journal.pone.0224365

---

## Author Comment (AC1)

Dear Editor,

Below, the contents of community comment 1 (CC1) by Seleshi W Gudeta are provided in black text and our responses are marked in blue text.

Dear Seleshi W Gudeta,

Thank you for taking the time to review our manuscript. We will address the comments and revise the paper accordingly.

Dear Editor,

**Comment 1.** This is a very useful work and I congratulate the authors for taking the initiative.

Response 1: We are grateful for the positive comments indicating that the work is very useful.

**Comment 2.** I have the following concerns, which I believe the authors will address for this work to be useful.

(1) My main concern relates to the discrepancy between the map they produced in Figure 7 and the Soil Atlas of Africa (see Jones et al., 2013), which is currently the authoritative reference material. For their map to be useful, it is important to reconcile with the map and wherever discrepancies exist it will be helpful to explain.

**Response 2:** We thank Seleshi W Gudeata for the comments. The following are our responses:

We acknowledge that the Soil Atlas of Africa is still useful to provide harmonisation and improvement, however, it is too general for diverse soil information users at local levels. It is derived from the Harmonized World Soil Database (HWSD) with expert-based modifications. The HWSD for East Africa, including Ethiopia, combines existing data/maps from the Soil and Terrain (SOTER) and SOTER-based soil parameter estimates (SOTWIS), while the soil map in SOTER has the following limitations:

- it is based on qualitative (polygon) maps, which were based on the previous maps.
- the SOTER soil nomenclature doesn't meet the present demand since it is based on FAO 1974 and FAO soil map of the world revised legend 1988 (reprint FAO-1990).
- since it is on a smaller scale, it depicts the dominant soil types from a larger area coverage and masked important soil units which would have been reported if a larger scale had been used. For example, in the HWSD, in the delineation of a given soil type, only the major one is reported, while up to 9 soil types coexist in each delineation.

- the geographic location of the dominant and associated soil types is not defined as it is based on a qualitative approach

Conclusion: The existing spatial soil information of Ethiopia is based either on a conventional/traditional qualitative approach using the mental model for extrapolation or quantitative/ digital soil mapping with limited unevenly distributed profile observations. Currently, we do not have a consistent spatial soil types information for Ethiopia, which necessitated the development of EthioSoilGrids 1.0.

On the other hand, the development of the EthioSoilGrids 1.0 is based on the following state-of-the-art techniques and procedures:

- it is based on rigorous quantitative spatial predictive model (Machine learning) that combine information from soil observations with environmental variables/covariates and remote sensing products.
- the mapping of soil types is based on the quantitatively defined probability of occurrence of each reference soil group (RSGs) per modelling window (250 meters).
- it is based on a much larger number of soil profile observations than any other soil mapping initiatives layering Ethiopia.
- the process of its development involved soil profile-based harmonization and translation to IUSS WRB 2015.
- it followed a hybrid approach, i.e., a combination of digital soil mapping, and expert validation of the soil types and their spatial patterns for generating consistent and updatable national spatial SoilGrid.

Therefore, given the above differences, in the approaches followed, scale, data source, etc, one should expect the difference between the Soil Atlas of Africa and the EthioSoilGrids 1.0. In other words, the latter is developed not to match the former, but to come up with improved and quality soil information, an objective fully achieved. Consequently, we are not surprised that the two products do not coincide since that was the assumption when the work was initiated. By the way, this is not the first report on Ethiopian soils' information showing such discrepancies as compared to the global products; for example -the spatial soil grids layering Ethiopia based on digital soil mapping techniques (e.g., SoilGrids, 2017) a similar approach followed in the preparation of EthioSoilGrid 1.0, reflected differences in RSGs area coverage.

**Comment:** Below is some of the discrepancies:

**Comment 2.1**: Cambisols are represented by a small proportion of the area in isolated pockets of Ethiopia according to the Soil Atlas of Africa. On the other hand, in this manuscript, Cambisols are the top-ranked in Figure 8. The explanation given for this in the manuscript is unsatisfactory.

**Response 2.1**

Cambisols' most abundance is acceptable, because Cambisols are developed in areas where pedogenetic development is slow (i) because of continuous erosion, but is in equilibrium with the weathering process, or continuous erosion and depositional cycles are common. As the result, they covered significant parts of the highlands of Ethiopia at the foot-slopes of undulating mountainous or hilly terrains, where erosion and weathering processes are in equilibrium, or erosion and deposition

cycles are common. (ii) because of low precipitation, or weathering-resistant parent materials. In this case, Cambisols occur in the large area of the lowlands of Ethiopia on weathering-resistant calcareous limestone, and on colluvial and alluvial deposits, where precipitation is low.

It is worth noting that the total number of profile observations per reference soil group (RSGs) in which Cambisols ranked third (with n=2219) following Luvisols (n= 2,229) and Vertisols (3,935). In fact, in some of the existing conventionally made country-wide legacy soil maps of Ethiopia, Cambisols were reported to cover e.g., 21% and 16% of the land mass of Ethiopia.

**Comment 2.2:** Areas bordering Djibouti and Eritrea that are predominantly covered by Leptosols (according to the Soil Atlas of Africa) are now covered by Fluvisols according to this manuscript. Many of these mountainous areas are not expected to have Fluvisols because Fluvisols naturally form in fluvial, lacustrine or marine deposits and periodically flooded areas.

**Response 2.2.** Yes, as noted by Seleshi W Gudeta, Pedogenetically Fluvisols are developed on flood plains, riverbanks, and lacustrine deposits. Since the areas bordering Djibouti and north-eastern lowlands (Afar and Somali lowlands) are under the influence of floods; where deposits from Awash, Wabishebele and Genale rivers are frequent, the predominance of Fluvisols is expected. Note that Leptosols are well represented on the volcanic mountains of Fantale, Boseti Guda and Ziqualla in the Awash valley, volcanic hills of the Afar lowlands, and the eastern escarpment of the central and northeastern rift valley, which are situated in these areas.

**Comment 2.3:** Areas in eastern and south-eastern Ethiopia bordering Somalia that are predominantly covered by Calcisols and Gypsisols (according to the Soil Atlas of Africa) have a continuous cover of Cambisols and some Fluvisols according to this manuscript. That cannot be possible.

**Response 2.3:** On comments about the formation and distribution of Cambisols and Fluvisols, we addressed the above in responses 2.1 and 2.2.

EthioGridSoil 1.0- is based on measured point observations collated from these areas after excluding RSGs with less than thirty observations including Gypsisols which had only 11 profiles. In this case, Gypsisols are excluded from mapping. Regarding Calcisols, as indicated by Seleshi W Gudeta, the probability of occurrence map (Figure C1 of Appendix C) depicts Calcisols dominantly occurring in eastern and south-eastern Ethiopia, bordering Somalia. However, when the relative abundance of RSGs per modelling window is assessed, Calcisols' area coverage as the dominant soil type as depicted in Figure 7, is the 7th most abundant soil in Ethiopia.

By the same token, in the polygon-based soil mapping like Soil Atlas of Africa, where a polygon is mapped as one soil unit does not mean that the polygon 100% represents that specific soil unit, but it also contains associations which are not depicted as dominant. Further, both the dominant and association geographic locations are not defined and hence do not directly indicate the specific location of each soil type.

**Comment 2.4**: Areas in north-western Ethiopia bordering Sudan that are predominantly covered by Nitisols, Luvisols and Alisols (according to the Soil Atlas of Africa) have almost a continuous cover of Vertisols according to this manuscript. That also does not make sense given that Vertisols form in depressions and level plains.

**Response 2.4:**

The north-western part of Ethiopia bordering Sudan from the Tekeze river (Humera area) down to the Baro basin is dominated by Vertisols while Luvisols and Nitisols intermingled before these two RSGs become dominant in relatively near distance/landscapes. The proportion of each soil type varies across the landscape. However, both the quantitative and qualitative assessments in those areas showed good agreement at this level of accuracy while the occurrence probability of each RSG is reported.

**Comment 2.5:** Andosols were shown in Eastern Ethiopia where they are not expected to occur (Andosols are formed from volcanic ejecta) and are common in the Rift Valley. Their occurrence outside is uncharacteristic.

**Response 2.5:**

Andosols are confirmed to occur outside the rift valley especially in the highland volcanic regions in the presence of organic matter. In Ethiopia, Andosols occur along the rift valley and on highlands for Examples on Bale mountains, Siemen Mountains (RasDashen), Choke Mountain, Abune Yosef Mountain and other mountains of the country. Below are some of the published references for confirmation:

Reference:
Assen, M., and Belay, T. 2008. Characteristics and classification of the soils of the plateau of simen mountains national park (smnp), Ethiopia.
Belay ,T.1995. Morphological, physical and chemical characteristics of Mollic Andosols of Tib Mountains, Central Ethiopian Highlands. SINET: Ethiop. J. Sci. 18 (2): 143–169.
Simane, B., Zaitchik, B.F, and Mutlu, O. 2013. Agroecosystem Analysis of the Choke Mountain Watersheds, Ethiopia" Sustainability 5, no. 2: 592-616. https://doi.org/10.3390/su5020592.
Gebrehiwot, K., Desalegn, T., Woldu, Z., Sebsebe, D., and Ermias, T.2018. Soil organic carbon stock in Abune Yosef afroalpine and sub-afroalpine vegetation, northern Ethiopia. Ecol Process 7, 6 (2018). https://doi.org/10.1186/s13717-018-0117-9.

In our study, the overall occurrence and the relative position of each of the reference soil groups along the topo sequence and its association with other RSGs agree with previous works and pedological expected/established schematic sequences. However, there were cases where the RSGs' position along the topo-sequence and association with other reference soil groups required further investigation, which was not adequately captured and explained in this study. This might be attributed to the positional accuracy of legacy point observations, modelling approach, and most importantly the level of details and scale/resolution of the environmental variables used in this study. For clarity, we will specify areas that require explanation arising from the above-stated likely reasons.

**Comment 3:** The colour coding in the map is confusing. For example, Acrisols, Cambisols and Leptosols were shown with colours that look alike. For this map to be useful it will be good if it is done with the same colour coding as the Soil Atlas of Africa and the Harmonisation of the soil map of Africa described in Dewitte.

Jones, A., Breuning-Madsen, H., Brossard, M., Dampha, A., Deckers, J., Dewitte, O., Hallett, S., Jones, R., Kilasara, M., Le Roux, P., Micheli, E., Montanarella, L., Spaargaren, O., Tahar, G.,

Thiombiano, L., Van Ranst, E., Yemefack, M. and Zougmore, R. (Eds.), (2013). *Soil Atlas of Africa. European Commission*, 176 pp., European Commission Luxembourg. DOI: 10.2788/52319

Dewitte, O., Jones, A., Spaargaren, O., Breuning-Madsen, H., Brossard, M., Dampha, A., Deckers, J., Gallali, T., Hallett, S., Jones, R., Kilasara, M., Le Roux, P., Michéli, E., Montanarella, L., Thiombiano, L., van Ranst, E., Yemefack, M. and Zougmore, R. (2013). Harmonisation of the soil map of Africa at the continental scale. *Geoderma* 212: 138-153. ODI: 10.1016/j.geoderma.2013.07.007.

**Response 3:**

As commented, we will address the colour coding and ensure distinct contrast among RSGs.

**Comment 4:** My appeal to the authors is to compare the soil profile data used for creating the map with the data used for the Soil Atlas of Africa.

**Response 4:**

See the preceding responses!

**Comment 5:** It is also important to check whether imbalances in sample sizes among soil types (e.g., preponderanc of vertisols and fewer Gypsisols) has influenced the analysis.

**Response 5:**

Kindly note that again Gypsisols are confirmed to occur based on the point profile observations but excluded from the modelling and not mapped in EthioSoilGrids version 1.0 product. However, as admitted in Line 441 to 444 of the manuscript, balanced datasets are ideal for modelling and mapping but the effect of datasets with uneven class along with various data treatment (pruning) techniques are recommended for future studies. The reason for this was that as we know there are different unbalanced categorical data treatment techniques targeting majority or minority classes leading to different predicted map accuracy and different overall, producers and users' accuracy.

---

## Author Response (AR2)

**Reference Soil Groups Map of Ethiopia Based on Legacy Data and Machine Learning Technique: EthioSoilGrids 1.0**

Author's response: **Ashenafi Ali et al**

**RC1- Skye Wills**

We thank Skye Wills (RC 1) for taking the time to review our manuscript. We respond to the issues raised as indicated below:

I commend the authors for this large and important effort and I appreciate the chance to review this work. This is a worthy effort that should be published and shared widely.

Response 1: Thank you for taking the time to review our manuscript and we are grateful for the positive comments.

I am very keen to explore the intersection between digital tool and expert knowledge in soil survey. However, reading this manuscript, I found myself with some additional questions and points of clarification needed. At numerous points, information was provided, but out of the order the reader might expect. This is at least partially due to the iterative nature of the project; but I found that some of the results were like part of the methods and some of the results read like conclusions. The repetition of information might cause a reader to skip sections and miss important pieces of information. I think with some additional explanation and minor edits, this paper will be ready for publication.

Response 2: Thank you for the comments. We improved issues related to redundancy, mix-up of statements in the methods, results and conclusions in the revised manuscript.

*We have considered the comments and revised the manuscript. Kindly, see sections 2.4.1; 2.4.3; 3.1; 3.2.1; 3.2.2; 3.2.3; 3.3; 3.4; and 4.0.*

Please find specific comments by line number:

Line 57: What number of profiles were used in the notable efforts referred to above (soilgrids 1 and 2)? How many of the thousands collected were included. This information would link the two parts of the intro – soil maps and soil profile collection.

Response 3: During legacy data collection campaign, over 20,000 profile data were collated (line 107). However, 14,742 profiles (Fig.4, line 265 to 267) were georeferenced with reference soil group naming. Following exclusion of five reference soil groups from the modelling, only 14, 681 profiles (line 112) were used for developing Ethio-Soil Grids v 1.0. In fact, some profiles data might have been dropped during the modelling process due to lack of data values with the corresponding covariate(s) as depicted in the confusion matrix. However, the global soil grids (1 and 2) development is based on the Africa soil profile database/global soil profile database in which only about 1,712 profiles (line 283) covering Ethiopia were used. These soil profile information are included in the development of EthioSoilGrid 1.0

Line 59: What do you mean that gridded spatial soil info is hardly available. Do you mean they were inaccessible, hard to use, incomplete? Please be explicit in explaining why the previous products were not adequate.

Response 4: We wanted to say that a national quantitative and spatially continuous predicted reference soil group/soil type map does not exist. We admit that hardly available is confusing and in the revised manuscript, it is revised by "does not exist". We explain why the previous products were not adequate in lines 48 to 69, as you noticed, especially in line 64. Further, we will revisit the statements.

*The statement has been changed as ….“Furthermore, country-wide quantitative and gridded spatial soil type information does not exist (Elias, 2016)”……*

Line 64: This paragraph makes more sense to me prior to the previous paragraph – to line 59.

Response 5: Thank you for this feedback. Your concern regarding line 59 will be addressed as indicated in response 4.

Line 71: What do you mean by improved?

Response 6: We wanted to mean we will develop an improved 250m soil grid map, which is more accurate as compared to the available global and regional soil grids.

Line 121: this is the accuracy of the profile data. Figure 2. What is Data Ecosystem Mapping? Does this include getting the metadata for each profile correct according to the covariates?

Response 7: The data ecosystem sketch is an effort to summarise the efforts involved starting from data sourcing to single standardised database. Data ecosystem mapping is the activity conducted to locate which data is available including the type of format and the level of completeness. It included getting metadata of each profile data. Harmonization of the coordinate reference system according to the covariate and different soil classification systems was worked out in the "Standardization phase" of the process.

Line 152: Are the terrain variables used listed anywhere…………. I see I think this paragraph is confusing as many of the details I was looking for are in the next paragraph. I recommend creating one paragraph or a separate climate and topography paragraph. Please list the DEM derivatives.

*All the covariates have been listed and key for abbreviations has been included as footnote under Figure 5.*

Response 8: All the variables including DEM variables listed in Appendix B. We will consider creating separate paragraphs for climate and topography.

Line 176: Did you consider evaluating your covariates for correlation and limiting the number used? Why or why not?

Response 9: We selected covariates representing the soil-forming factors based on expert knowledge and a review of the literature. We used near zero variance analysis to reduce variables that are not contributing to the RSG modelling and mapping. We didn't test covariates for correlation because we opted to include any covariates as long as it contributes to the prediction. This is in line with the suggestion by Helfenstein et al (2022) who stated that Ensemble decision tree models are robust
against highly correlated data and we consider prediction accuracy more important than model
interpretability. Based on the suggestion of the reviewer, however, we have explicitly indicated that
correlation between the covariates is not done in the analysis.

Helfenstein, A., Mulder, V. L., Heuvelink, G. B., & Okx, J. P. (2022). Tier 4 maps of soil pH at 25 m
resolution       for       the       Netherlands.       Geoderma,       410,
115659. https://doi.org/10.1016/j.geoderma.2021.115659

Line 179: this paragraphs seems more introductory and not part of explaining your process.

Response 10: Thank you, we revised it accordingly.

*The statement has been removed.*

Line 194: Are you saying previous studies have used this technique? I think you could
eliminate this sentence.

Response 11: Thank you this is deleted.

Line 199: were optimized how? Is there a metric you were evaluating? Does the Caret
package give you some sort of evaluation?

Response 12: "expand.grid" function in Caret package was used  to create a set of different tuning
features while training the model. The three tuning parameters for Ranger method in Caret package
are mtry, splitrule, min.node.size. Generally this function is used to tune the parameters in modelling
in an automated fashion, as this will automatically check all the possible tuning parameters and return
the optimized parameters on which the model gives the best accuracy.

Line 202: Did you state how you separated the training and testing sets and what the 'new'
dataset is? You should define those sets, and how they were selected and used.

Response 13: The function "createDataPartition" was used to create balanced splits of the data. As the
y argument (response variable) to this function is a factor, the random sampling occurs within each
class and preserves the overall class distribution of the data.

Line 224: typo '-runto'

should have a space '-run to'

Response 14: Thank you. Corrected accordingly..

Line 254: Consider something more definitive and eliminate 'the results suggest'. I think
these are straightforward results that need no wiggle words like 'suggest'.

Response 15: We will correct it as commented. *Corrected as suggested/commented*

Line 255: I am not sure the word 'museum' is what I would use here. Perhaps 'display' or
'diversity' is more appropriate?

Response 16: Thank you and revised accordingly.

*The statement has been revised as "…. a land of soil diversity…."*

Line 268: Is this section not part of the methods? This describes how you collected and
evaluated profiles, which is covered earlier.

Response 17: In this section, we are describing the spatial density of the new database, which is one
of the key results of this work. In doing so, we present these results by comparing with existing and
previous databases used for developing similar soil group maps. We think these are appropriate results
to be presented in this section. Therefore, we do ask the kind understanding of the reviewer to allow
us to maintain this description as it is and where it is.

Line 323: This is a great description of the setting and climate; but I think it might fit better
in the methods or introduction. Figure 6. My preference is to rename the covariates or list the
abbreviations in the figure captions. It is cumbersome for the reader to have to toggle
between this figure and an appendix.

Response 18: In this section, the effort is to explain the different covariates that are important in
predicting the soil type. In order of their importance, we tried to explain what would be the reason
why these factors are important in defining the soil type based on our experience and existing
literature. That is what and why the climate is detailed in this section. Based on your comment, we
added the description of the variable in the caption of figure 6 for easy referencing.

*List of variables has been added in the caption of figure5.*

Line 357: could the low influence of lithology have anything to do with WRB class breaks
and how they intersect with the scale of parent material variability?

Response 19: It is the relative importance which is low, and may be related to the use of a coarse-
scale  and less detailed lithology map, which may not sufficiently capture the spatial variability of the
parent materials.

Line 361: can you take mtry and the comma out of this sentence, does it still mean the same
thing?

Response 20: we revised this for clarity. It is basically mtry = 20, split rule = extra trees and minimum
node size = 5.  For better clarity, the sentence will be revised. See also Response 12.

Line 362: Did you test the accuracy of previous maps or find other reported accuracies of
maps from the area (not just general averages)?

Response 21: We didn't test the accuracy of previous maps rather we used the reported accuracies
from published sources.

Line 375: I am very curious what the accuracy of Global Soil Grids is using your updated soil
profiles. Without that information, it is difficult to know how successful this effort using
expert knowledge has been.

Response 22:  Here we wanted to communicate that qualitative assessment of spatial patterns was not
done for SoilGrids 2017 which considers soil type mapping. This is to indicate similar accuracy might
lead to different spatial patterns and hence expert-based qualitative evaluation is of paramount
importance.

Line 401: the portion of this paragraph dealing with landscapes/top-sequeces belongs with the paragraph below (line 409) focused on topo-sequences.

Response 23: Thank you for the observation, this is revised accordingly.

Line 426: Are the soil qualities (I think you mean properties) transitional or are the covariates transitional (or both?).

Response 24: yes properties, properties transitional implies it is because of the covariates/soil forming factors and hence we can say both.

 Line 441: I think this is an 'and' not a 'but'. Did you consider adjusting you training dataset for more balanced set of soil profiles?

Response 25: For randomly sampling and splitting the dataset into training and testing set, we tried different set.seed values to ensure inclusion of each RSGs in both splitted sets and better accuracy. See also Response 13

Line 445: this paragraph read very much like a concluding statement, was that the intention?

Response 26: Thank you - we have revised accordingly. Some parts of this paragraph are revised and maintained there. The other descriptions which look like conclusions are taken to the conclusion section.

*This comment has been addressed under the respective sections in the revised manuscript!*

Line 458 – Section 458. It would be much more powerful to compare the expert evaluation of this map vs. the expert evaluation of previous maps. Was any re-evaluation done after re-running the model. Did the output from the tests change throughout the process? Were the scales used to evaluate by experts useful to the scale of your model?

Response 27: After re-running the model, about ten soil scientists and geospatial experts re-evaluate the output using 20-25 districts. Further, the geospatial and soil experts checked the raster map of the RSGs in GIS environment to ensure areas with no concern before re-running the model are kept the same or changes are acceptable. The quality of input data (profile data, covariates, mask layer) was assessed to improve the overall accuracy. As a general working norm, the expert's qualitative assessment was set to consider the representation of mappable soil types at the target resolution/scale.

**RC2- Sky Wills**

Dear Sky Wills (RC2),

Kindly please refer to our response (AC6) to RC1, as both RC1 and RC2 are the same.

Kind regards,

Ashenafi Ali (on behalf of the co-authors)

**RC3- Anonymous**

We thank anonymous Referee #2 for valuable suggestions and comments, which have greatly contributed to the enhancement of our manuscript. Our responses are provided in each comment and suggestion by the referee:

**Overall evaluation:**

● I feel that the paper is a great effort by the authors to draw together a set of soils data for Ethiopia and improve the spatial resolution of the mapping. I think just pulling together the data set is a big achievement.

**Response 1:** Thank you for the positive feedback and compliments on our work

● However, I feel the paper lacks a critical evaluation of the results and of the subsequent learning and recommendations that could be made. To do this it needs an assessment of where the modelling worked well and where it didn't and explanations of why these results may have occurred.

*We have considered the comments and revised the manuscript. Kindly, see sections: 3.1; 3.2.1; 3.2.2; 3.2.3; 3.3; 3.4; and 4.0.*

**Response 2:** Thank you for the comment. The modelling accuracy was assessed based on the standard cross-validation technique that involves the overall map accuracy. It is a resource and time-demanding (which also was not the scope of the present study) to consider model-free and design-unbiased accuracy assessment which is believed to be achieved with probability sampling, while taxonomic correctness is one of the key determinant factors to be considered in such class/Reference Soil Groups (RSGs) mapping.

Digital soil mapping (DSM) product users have indicated critical concerns to what degree DSM products represent the actual soil landscape spatial patterns, as similar/close quantitative accuracy statistics might show different soil class spatial patterns. To address this concern, we employed an expert-based qualitative assessment of the model output. This technique was used to complement model-based accuracy assessment and confirm/indicate where the modeling specifically worked well and where it didn't. This was implemented by a panel of senior soil specialists/pedologists checking the map based on objectively selected geographic windows across Ethiopia, representing different agroecological zones known to have diverse soil occurrences, and familiar to the panel of experts. Accordingly, the outcome of the evaluation which is an indicator of the model performance across geographic windows presented interms of aggregated ratings (lines 229 and 230): 1. confirmed with '*no concern*', 2. confirmed with "*minor concern*", and 3. confirmed with '*major concern*'. However, we accept the comments and we will elaborate on the findings of the qualitative evaluation as per pedological-based interpretations/assessments both in the examined geographic windows and prominent contrasting landscapes of Ethiopia.

 To provide some reflection on the basis of spatial windows, for instance,  in the northeastern lowlands of Ethiopia, mainly along the "Denakil" depression, it is observed that the model overestimated Fluvisols; and confused Fluvisols with Vertisols. Further, mainly Solonchaks, believed to be peculiar features of that particular landscape and Leptosols are under-represented. In some parts of the southeastern lowlands of Ethiopia, Calcisols spatial distribution is under-represented and Cambisols were overestimated. The modeling didn't work well in these cases which may be attributed to the low number of soil profile observations (Figure 5) in those areas. This implies that we need additional soil profile observations. The above discussion will be added in the revised version under the new heading **3.4. Evaluation of results and future direction.**

*Section 3.4 has been added:*

**3.4 Evaluation of results, limitations and future direction**

[revised manuscript text omitted]

● I think the discussion of the maps with experts is a really useful way of validating the maps and more could be made of the results of these discussions.

*We have considered the comments and revised the manuscript. Kindly, see sections: 2.4.1; 2.4.3; 3.1; 3.2.1; 3.2.2; 3.2.3; 3.3; 3.4; and 4.0.*

**Response 3:** We accepted the comments, we will add more soil-landscape-based elaborations (kindly see Response 2) based on examined geographic windows and well-known national spatial patterns, as the team involves a panel of senior soil surveyors/experts/pedologists who have been involved in many soil survey and mapping missions across a mosaic of Ethiopia's landscapes.

● There needs to be a discussion about where results are unexpected/expected and how that links back to figure 5 and the availability of the input soil profile data and covariates in different areas.

*Through discussion, incorporating comments and suggestions have been included in the revised manuscript. Kindly, see sections: 3.4 and 4.0.*

**Response 4:** Thank you for this comment, we will address it (kindly see also Response 2). There are areas where fewer soil observations (explained in lines 285 to 287) and sparse geographical coverage affect the modelling performance. This was observed and reported by the panel of experts zoomed-in assessment across areas labelled as 'minor' and 'major' concerns and across some landscapes such as in the eastern lowlands. Besides, geographic coverage of quality input soil profile data, adequate representation of the feature space could affect the model performance. Sometimes given the covariate issue and examining spatial details relatively similar, some unexpected spatial patterns might be due to issues related to the adequacy of representing the feature space. In addition, the granularity, level of detail and quality of the covariates towards the model performance will be further elaborated, in such a way as to highlight areas that are worth consideration for future similar studies and efforts to improve the map accuracy.

● The paper needs to highlight what we can learn from mapping in Ethiopia for mapping in similar landscapes. If this can be added I think it would be a really valuable addition to e DSM literature.

*Further, through discussion incorporating comments and suggestions have been included in the revised manuscript. Kindly, see sections: 3.4 and 4.0.*

**Response 5:** One of the key insights gained from this study is the critical role of collating existing soil profile data. It is important to recognize that conducting repetitive soil characterization and classification exercises or an effort to update existing legacy soil maps through new soil survey campaigns can be both costly and time inefficient. Similarly, for countries like Ethiopia which are very vast and characterized by diverse soil forming factors and soil resources, a conventional mapping approach would be much more resource and time-demanding. Therefore, it is imperative to explore alternative approaches that maximize the utilization of available and optimal soil profile data and digital soil mapping techniques which the paper aims to address.

In addition, addressing the issue of data standardization within data collation methodologies is of utmost importance. By establishing standardized data collection practices, we can ensure the compatibility and comparability of collated data for effective utilization in digital soil mapping (DSM) models throughout Africa. The paper emphasizes the significance of implementing data collection standards and practices in Ethiopia and other Sub-Saharan African regions. This will enable the generation of a sufficiently large number of observations, which are essential for developing data-driven DSM models and other precision agronomy applications.

It is essential to note that the recommendations presented in this paper extend beyond Ethiopia's borders and hold relevance for other countries in Sub-Saharan Africa. These recommendations provide valuable insights and guidance for the adoption of standardized data collection practices across the region. By embracing these recommendations, researchers and practitioners can ensure the generation of high-quality data, thereby facilitating the development of robust and effective DSM models and precision agronomy approaches. Some of these learnings will be added and discussed in the revised manuscript.

*Further, through discussion incorporating comments and suggestions have been included in the revised manuscript. Kindly, see sections: 3.4 and 4.0.*

*"Up-to-date soil resource spatial information is critically missing at a required scale and extent in Ethiopia. As a result, resource management strategies miss their targets. Furthermore, the absence of such data at a required resolution and extent, forced decision support tool developers to pick and use the data they can access and afford. As a result, model outputs appear more site-specific or representation becomes homogenous over the very heterogeneous landscapes that exist in reality. On the other hand, in large areas and complex landscapes such as Ethiopia, it is very difficult to address the demand for reasonably accurate and detailed soil-type maps using a conventional approach due to the costs involved, and resources and time it requires. For instance, given the vastness of the country and heterogeneous landscapes, a new conventional soil survey mission requires at least 170,000 profile point observations to map the entire terrestrial land mass of Ethiopia at a scale of 1: 250,000 with at least 1 observations per square centimetre. Moreover, the soil profile data requirement definitely could have been much higher as we increase the scale of mapping and density of observations. In the present study, machine-learning techniques combined with expert input were implemented to produce a countrywide soil resource map of Ethiopia at reasonably higher accuracy, less time and cost than that of conventional methods. In addition, rescue, compilations and standardization of about 14,681 geo-referenced legacy soil profiles that can be included in the National Soil Information System (NSIS) of Ethiopia and the World Soil Information Centre will support future national, regional and global DSM efforts. The approach used demonstrates the power of data and*

*analytics to map the soil resources of Ethiopia and the output is an exemplary use case for*

*similar digital content development efforts in Ethiopia and beyond.*

*Moreover, in this study the quality monitoring processes and methods were followed to filter*

*dubious soil profiles, and soil classification and harmonization protocols. Then after, the*

*study followed a robust modelling framework and generated new insights into the relative*

*area coverage of WRB RSGs of Ethiopia. In addition, the study provided coherent and up-to-*

*date digital quantitative gridded spatial soil resource information to support the successful*

*implementation of various digital agricultural solutions and decision support tools (DSTs)."*

**Specific queries:**

● Could the resolution of the input data explain why the results may not be as expected in
certain areas?

**Response 6:** Yes, among other factors, if we have separately examined the effects of the covariates, the spatial resolution and level of detail could contribute to why the results are unexpected in certain areas. For instance, within the given spatial level of examination, the sequence of some RSGs showed different patterns which could be captured by better resolution parent material map in the SCORPAN model. We will highlight this issue in the revised manuscript.

● In the discussion of the confusion matrix (Table 1) the authors could look at where
there are large differences between soils pedologically and where a miss mapping of soils
might lead to different management decisions in areas.

**Response 7:** Thank you for raising this issue and for the comments. In the confusion matrix (Table 1), the quantitative classification errors (omission and commission errors) need to be interpreted/checked in terms of the soil's pedological similarity/differences which is commonly called 'taxonomy distance'. It is such an evaluation that will add value to interpreting the errors from producers' and users' perspectives and check areas of concern to implement management decisions. In soil class mapping where classification accuracy is represented by a confusion matrix, literature indicated, it is likely that not all errors are equally serious. Some errors are more serious than others in terms of soil properties, soil- forming process, ease of map making and application of the map. For instance, from the user's perspective, Vertisols predictions were distributed to incorrect Leptosols and Nitisols classes which implies leading to significantly different management decisions in terms of soil depth, aeration, and acidity. The same applies to miss mapping of Arenosols as Luvisols and Vertisols. The miss-mapping interpretation needs to be supported based on the soil's taxonomic distance, which determines class similarity and dissimilarity determining different management decisions and hence, implying, fractional recognition needs to be given to some incorrect allocations represented in the confusion matrix.

- ● The paper mentions a rerun of the modelling after the workshop. Can the authors explain what was changed to improve the results between the 2 runs and which versions of the runs are presented in this paper.

**Response 8:** After re-running the model, about ten soil scientists and geospatial experts (lines 242 and 243) re-evaluated the output using districts selected based on the feedback from the first review, which was mainly on areas where there was "minor" and "major" concerns. For instance, in areas where Vertisols, Fluvisols, and Leptosols were reported to be overestimated, improvements were observed. Further, underestimated RSGs (Alisols, Solonetzs, Planosols, Acrisols, Lixisols, Phaeozems, and Gleysols) showed slight area coverage and pattern improvements. However, the total area for Leptosols and Cambisols increased from the first run due to the partial exclusion of the mask layer used in the first round modeling effort. The mask layer used in the first run was criticized for quality issues as it excluded significant soil areas and its limitation to capturing non-soil areas such as rock outcrops/rocky surfaces, salt flats, swamps and sand dunes across the different landscapes. Nevertheless, the spatial patterns of these soils occurring across previously considered "non-soil areas'' were examined by the panel of experts. In parallel, geospatial and soil experts checked the raster map of the RSGs in the GIS environment to ensure areas with 'no concern' before re-running the model are kept the same or changes are accepted by the panel of experts. The map from the second run is presented in this paper.

- ● I think its structure needs some thought specifically. The results of the validation described in section 2.4.2 need to be part of the results rather than the methods.

*Sections 2.4.2 and 3.3 have been revised and improved.*

**Response 9:** Thank you for the comment. In section 2.4.2. we presented how we did the qualitative validation procedures (i.e. expert evaluation) and the outcome of this process is presented in the result section (sec 3.3). We thought this flow was much easier to follow the paper. Therefore, we kindly ask the reviewer to allow us to maintain the current structure of these sections.

**Points of clarification:**

● **Line** 59: What is meant by "hardly available"

**Response 10:** As elaborated for Referee 1 (See Response 4 of AC 7) we wanted to say that a national quantitative and spatially continuous predicted reference soil group/soil type map does not exist. We admit that hardly available is confusing and in the revised manuscript, it has been revised by "does not exist".

● **Line** 113: What criteria were used to define if a profile is complete and clean?

**Response 11:** The criteria used were basic profile information/data required for classification of RSGs. For clarity, the statement will be amended as: ..... cleanness, i.e., profile points with basic data/information for classification of RSGs.

● Line 223: How were the polygons for review selected?

**Response 12:** In order to represent every part of the country, the polygons/geographic windows for qualitative assessment were purposely selected by a panel of senior soil specialists/pedologists/soil surveyors before breakout sessions and proceeded to group works. The revised version has been be updated by adding the phrase "purposely". The experts were drawn from different corners of the country and had been involved in different soil survey missions across Ethiopia. Hence, each suggested geographic window was debated and agreed upon based on soil diversity, contrasting/unique soil-landscape relations, availability of familiar experts in the panel, and agro-ecological zone coverage.

● Line 233: How are the authors looking to improve the version of the map from the first version?

*Kindly, see sections: 3.4 and 4.0.*

**Response 13:** Thank you for raising this issue. The first version of the map will be improved by ensuring additional input profile data from under-represented geographic and feature spaces, and covariates with improved resolution, quality and level of detail including through the implementation of different covariate selection procedures. Application of a robust modeling technique that accommodates neighbourhood size and connectivity analyses requires due consideration by future studies. It is also recommended to implement unbalanced data treatment and de-clustering techniques to overcome issues likely to arise from class imbalances and biased datasets in such kinds of soil class/type mapping efforts. The above statement will be added in the revised version under the new section, 3.4. Evaluation of results and future direction.

*Kindly see section 3.4 in the revised manuscript "……. To improve the modelling performance, future studies could explore (1) adding data for under-represented geographic areas, land uses and covariate spaces, (2) opportunities to include other covariates ( parent material and management) that could capture variability of the country heterogeneous landscapes, (3) dimension reduction of covariates (4) use of remedial measures for imbalances in sample sizes, (5) comparing different cross-validation methods, (6) use of an ensemble modelling approach and/or robust modelling technique that accommodates neighbourhood size and connectivity analyses, (7) use of better resolution/quality mask layer to segregate non-soil areas ( rock outcrops, salt flats, sand dunes and water bodies) from mapping areas, and (8) implementation of quantitative and qualitative comparison of national, regional, and global legacy soil maps/soil grids with new DSM products in terms of how well DSM products represent soil geography. In addition , future digital soil mapping strategies in Ethiopia may require to consider new soil sampling missions in under-represented areas, adopt standard soil sampling, description guidelines and soil classification systems including soil physico-chemical and mineralogical analysis, and combine local soil nomenclature/classification systems with RSGs and develop a map of RSGs with qualifiers. At the moment the under-sampled and under-represented areas are the Somali region, the Denakil and the western and north-western border areas of Ethiopia (Figure 4). Regardless of these limitations and to the best of our knowledge the EthioSoilGrids v1.0 product we presented here provides the most complete soil information available for Ethiopia."*

● Line 247 – 253: Do the number of samples used represent what would be expected in terms of areas of specific soils in Ethiopia or are the input data biased to specific land cover or soil types.

**Response 14:** In general, ignoring the temporal resolution, i.e., from the 1970s to the 2020s, the number of samples is expected to cover areas of important agroecological zones and land use/covers. However, in terms of areas of specific soils of Ethiopia, while the $1^{st}$, and the $2^{nd}$ largest input data were from Vertisols and Luvisols, their relative area coverages were in $3^{rd}$ and $6^{th}$ positions, respectively. This bias might have happened because of the soil survey interests. For example, many surveys focused on Vertisols and Luvisols for the purpose of agricultural intensification/mechanization and irrigation in areas where these soils are situated. This signifies the need to focus on future soil data collection to consider soils with fewer input data compared to their relative area coverage. Moreover, this study utilizes the most extensive soil profile observation data available to date for the generation of a comprehensive soil-type map of Ethiopia. Despite the inherent uncertainties associated with data representation, this is the first significant endeavor based on such a large-scale observation effort. This description will be added to the revised version under the new section 3.4. Evaluation of results and future direction.

● Line 274-278: Do the authors see a difference in the quality of the results where they had an increased density of input profiles?

**Response 15:** In general yes, but not in all the cases, for instance, based on geographic and feature space coverage and RSGs diversity.

● Figure 6: Add an axis label to the X axis

**Response 16:** Thank you for the comment. We will label it.

● Line 409-418: The authors need to discuss in more detail the reasons why certain points in the topographic sequences do match other work and where they don't and offer potential explanations of why.

**Response 17:** Thank you; we will elaborate further as suggested.

*Kindly, see sections: 2.4.2; 3.2.2; 3.2.3; 3.3; 3.4 and 4.0.*

*Elaborated as:*

*"However, in some cases, the RSGs' position along the topo-sequence and association with other RSGs require further investigation.  The observed disparities might be attributed to the positional accuracy of legacy point observations, modelling approach, and most importantly the level of detail and scale/resolution of the environmental variables used in this study. We used the currently available coarse resolution national geological map and hence soil parent material might be inadequately represented in the model, which probably resulted in irregular RSGs sequences. For instance, the main driving factors to establish and explain soil-landscape variability in May-Leiba catchment of northern Ethiopia were geology (soil parent material) and different mass movements (Van de Wauw et al., 2008). These factors led to Cambisols– Vertisols catenas on basalt and Regosols–Cambisols–Vertisols catenas on limestone formations. Similar studies identified parent material strongly determines the soil type (e.g. Vertisol, Luvisol, Cambisol) (Nyssen et al., 2019). In general, in areas where there is complex soil diversity and distribution of soils, one of the most important parameters is to identify parent material including effective techniques to capture and delineate mass*

*movement bodies, and human-induced soil erosion and deposition areas (Leenars et al.,*

*2020a; Nyssen et al., 2019; Van de Wauw et al., 2008).*"

● **Line** 428-435: This section assumes that the new soil grids that have been generated are better than the "soil grids" without explaining what the insight comes from the new modeling and why it's important. It would also be valuable if the authors could offer insight into which of the 3 reasons the results may be different.

The below statement has been added:

*"….This is mainly attributed to limited access to more local point data by regional and global modelling initiatives, unlike the present study which accessed a large number of legacy soil profile datasets….."*

*Kindly, see sections: 2.4.3; 3.2.2; 3.2.3; 3.3; 3.4 and 4.0.*

**Response 18:** Thank you for the comment. We will elaborate further. Kindly please note that we based our comparison on the reported map accuracies, implementation of expert-based qualitative assessment of spatial patterns, and number and distribution of input soil profile observations. We will elaborate more and recommend the need for quantitative comparisons of legacy soil maps (including "soilgirds") in terms of how well they represent soil geography. Hence, users will get insights into the applicability of various DSM products at different spatial scales and geographic windows.

● Line 441-444: Is it likely that the data used in this study are biased and can the authors offer a recommendation on what new data might be needed in which areas to improve the results.

*Kindly, see sections: 2.4.3; 3.2.2; 3.2.3; 3.3; 3.4 and 4.0.*

**Response 19:** Part of this query is addressed in the above (kindly see Reference 14). Keeping the temporal resolution constant, as the data source between the 1970s and 2020s, the input data are biased to specific land uses (cultivated/arable and grazing lands) and agroecological zones of Ethiopia (see lines 290 to 301). Hence, additional legacy data are required from less represented land uses such as forests, shrubs and bushlands. However, in some geographic areas such as the north and southeastern lowlands and in some agroecological zones where there is no/under-representation of input data, additional new data are required from more land uses.

• Lines 473-479 it is unclear whether the rerun version of the map is what has been presented
in the current paper whether that is something that is to follow. If it isn't presented can the
authors explain why not.

**Response 20:** Thank you for the comment, we will elaborate further. This query is addressed
in the above (kindly see Response 8). The map from the second run is presented in this paper.

**CC1- Seleshi W Gudeta**

Date: 27 June 2022

Dear Editor Subject: Response to interactive comment on our manuscript entitled: Ali et al.:
Reference Soil Groups Map of Ethiopia Based on Legacy Data and Machine Learning Technique:
EthioSoilGrids 1.0

By Ashenafi Ali et al.

Dear Editor,

Below, the contents of community comment 1 (CC1) by Seleshi W Gudeta are provided in black
text and our responses are marked in blue text.

Dear Seleshi W Gudeta,

Thank you for taking the time to review our manuscript. We will address the comments and
revise the paper accordingly.

Dear Editor,

Comment 1. This is a very useful work and I congratulate the authors for taking the initiative.

**Response 1:** We are grateful for the positive comments indicating that the work is very useful.

Comment 2. I have the following concerns, which I believe the authors will address for this work
to be useful.

(1) My main concern relates to the discrepancy between the map they produced in Figure 7 and
the Soil Atlas of Africa (see Jones et al., 2013), which is currently the authoritative reference
material. For their map to be useful, it is important to reconcile with the map and wherever
discrepancies exist it will be helpful to explain.

**Response 2:** We thank Seleshi W Gudeata for the comments. The following are our responses:

We acknowledge that the Soil Atlas of Africa is still useful to provide harmonisation and
improvement, however, it is too general for diverse soil information users at local levels. It is
derived from the Harmonized World Soil Database (HWSD) with expert-based modifications.
The HWSD for East Africa, including Ethiopia, combines existing data/maps from the Soil and
Terrain (SOTER) and SOTER-based soil parameter estimates (SOTWIS), while the soil map in
SOTER has the following limitations:

• it is based on qualitative (polygon) maps, which were based on the previous maps.

• the SOTER soil nomenclature doesn't meet the present demand since it is based on FAO
1974 and FAO soil map of the world revised legend 1988 (reprint FAO-1990).

• since it is on a smaller scale, it depicts the dominant soil types from a larger area coverage
and masked important soil units which would have been reported if a larger scale had been
used. For example, in the HWSD, in the delineation of a given soil type, only the major one is
reported, while up to 9 soil types coexist in each delineation.

• the geographic location of the dominant and associated soil types is not defined as it is
based on a qualitative approach

Conclusion: The existing spatial soil information of Ethiopia is based either on a
conventional/traditional qualitative approach using the mental model for extrapolation or
quantitative/ digital soil mapping with limited unevenly distributed profile observations.
Currently, we do not have a consistent spatial soil types information for Ethiopia, which
necessitated the development of EthioSoilGrids 1.0.

On the other hand, the development of the EthioSoilGrids 1.0 is based on the following state-of-
the-art techniques and procedures:

• it is based on rigorous quantitative spatial predictive model (Machine learning) that
combine information from soil observations with environmental variables/covariates and
remote sensing products.

• the mapping of soil types is based on the quantitatively defined probability of occurrence
of each reference soil group (RSGs) per modelling window (250 meters).

• it is based on a much larger number of soil profile observations than any other soil
mapping initiatives layering Ethiopia.

• the process of its development involved soil profile-based harmonization and translation
to IUSS WRB 2015.

• it followed a hybrid approach, i.e., a combination of digital soil mapping, and expert
validation of the soil types and their spatial patterns for generating consistent and updatable
national SoilGrid.

Therefore, given the above differences, in the approaches followed, scale, data source, etc, one
should expect the difference between the Soil Atlas of Africa and the EthioSoilGrids 1.0. In other
words, the latter is developed not to match the former, but to come up with improved and quality
soil information, an objective fully achieved. Consequently, we are not surprised that the two
products do not coincide since that was the assumption when the work was initiated. By the way,
this is not the first report on Ethiopian soils' information showing such discrepancies as
compared to the global products; for example -the spatial soil grids layering Ethiopia based on
digital soil mapping techniques (e.g., SoilGrids, 2017) a similar approach followed in the
preparation of EthioSoilGrid 1.0, reflected differences in RSGs area coverage.

**Comment:** Below is some of the discrepancies:

**Comment 2.1**: Cambisols are represented by a small proportion of the area in isolated pockets of
Ethiopia according to the Soil Atlas of Africa. On the other hand, in this manuscript, Cambisols
are the top-ranked in Figure 8. The explanation given for this in the manuscript is unsatisfactory.

**Response 2.1**

Cambisols' most abundance is acceptable, because Cambisols are developed in areas where
pedogenetic development is slow (i) because of continuous erosion, but is in equilibrium with the
weathering process, or continuous erosion and depositional cycles are common. As the result,
they covered significant parts of the highlands of Ethiopia at the foot-slopes of undulating
mountainous or hilly terrains, where erosion and weathering processes are in equilibrium, or
erosion and deposition cycles are common. (ii) because of low precipitation, or weathering-
resistant parent materials. In this case, Cambisols occur in the large area of the lowlands of
Ethiopia on weathering-resistant calcareous limestone, and on colluvial and alluvial deposits,
where precipitation is low.

It is worth noting that the total number of profile observations per reference soil group (RSGs) in
which Cambisols ranked third (with n=2219) following Luvisols (n= 2,229) and Vertisols
(3,935). In fact, in some of the existing conventionally made country-wide legacy soil maps of
Ethiopia, Cambisols were reported to cover e.g., 21% and 16% of the land mass of Ethiopia.

**Comment 2.2:** Areas bordering Djibouti and Eritrea that are predominantly covered by Leptosols
(according to the Soil Atlas of Africa) are now covered by Fluvisols according to this manuscript.
Many of these mountainous areas are not expected to have Fluvisols because Fluvisols naturally
form in fluvial, lacustrine or marine deposits and periodically flooded areas.

**Response 2.2.** Yes, as noted by Seleshi W Gudeta, Pedogenetically Fluvisols are developed on
flood plains, riverbanks, and lacustrine deposits. Since the areas bordering Djibouti and north-
eastern lowlands (Afar and Somali lowlands) are under the influence of floods; where deposits
from Awash, Wabishebele and Genale rivers are frequent, the predominance of Fluvisols is
expected. Note that Leptosols are well represented on the volcanic mountains of Fantale, Boseti
Guda and Ziqualla in the Awash valley, volcanic hills of the Afar lowlands, and the eastern
escarpment of the central and northeastern rift valley, which are situated in these areas.

**Comment 2.3:** Areas in eastern and south-eastern Ethiopia bordering Somalia that are
predominantly covered by Calcisols and Gypsisols (according to the Soil Atlas of Africa) have a
continuous cover of Cambisols and some Fluvisols according to this manuscript. That cannot be
possible.

**Response 2.3:** On comments about the formation and distribution of Cambisols and Fluvisols, we
addressed the above in responses 2.1 and 2.2.

EthioGridSoil 1.0- is based on measured point observations collated from these areas after
excluding RSGs with less than thirty observations including Gypsisols which had only 11
profiles. In this case, Gypsisols are excluded from mapping. Regarding Calcisols, as indicated by
Seleshi W Gudeta, the probability of occurrence map (Figure C1 of Appendix C) depicts
Calcisols dominantly occurring in eastern and south-eastern Ethiopia, bordering Somalia.
However, when the relative abundance of RSGs per modelling window is assessed, Calcisols'
area coverage as the dominant soil type as depicted in Figure 7, is the 7th most abundant soil in
Ethiopia.

By the same token, in the polygon-based soil mapping like Soil Atlas of Africa, where a polygon
is mapped as one soil unit does not mean that the polygon 100% represents that specific soil unit,
but it also contains associations which are not depicted as dominant. Further, both the dominant
and association geographic locations are not defined and hence do not directly indicate the
specific location of each soil type.

**Comment 2.4**: Areas in north-western Ethiopia bordering Sudan that are predominantly covered by Nitisols, Luvisols and Alisols (according to the Soil Atlas of Africa) have almost a continuous cover of Vertisols according to this manuscript. That also does not make sense given that Vertisols form in depressions and level plains.

**Response 2.4:**

The north-western part of Ethiopia bordering Sudan from the Tekeze river (Humera area) down to the Baro basin is dominated by Vertisols while Luvisols and Nitisols intermingled before these two RSGs become dominant in relatively near distance/landscapes. The proportion of each soil type varies across the landscape. However, both the quantitative and qualitative assessments in those areas showed good agreement at this level of accuracy while the occurrence probability of each RSG is reported.

**Comment 2.5:** Andosols were shown in Eastern Ethiopia where they are not expected to occur (Andosols are formed from volcanic ejecta) and are common in the Rift Valley. Their occurrence outside is uncharacteristic.

**Response 2.5:**

Andosols are confirmed to occur outside the rift valley especially in the highland volcanic regions in the presence of organic matter. In Ethiopia, Andosols occur along the rift valley and on highlands for examples on Bale mountains, Siemen Mountains (RasDashen), Choke Mountain, Abune Yosef Mountain and other mountains of the country. Below are some of the published references for confirmation:

**Response 3:**

As commented, we will address the colour coding and ensure distinct contrast among RSGs.

**Comment 4:** My appeal to the authors is to compare the soil profile data used for creating the map with the data used for the Soil Atlas of Africa.

**Response 4:**

See the preceding responses!

**Comment 5:** It is also important to check whether imbalances in sample sizes among soil types (e.g., preponderanc of vertisols and fewer Gypsisols) has influenced the analysis.

**Response 5:**

Kindly note that again Gypsisols are confirmed to occur based on the point profile observations but excluded from the modelling and not mapped in EthioSoilGrids version 1.0 product. However, as admitted in Line 441 to 444 of the manuscript, balanced datasets are ideal for modelling and mapping but the effect of datasets with uneven class along with various data treatment (pruning) techniques are recommended for future studies. The reason for this was that as we know there are different unbalanced categorical data treatment techniques targeting majority or minority classes leading to different predicted map accuracy and different overall, producers and users' accuracy.

**CC2- Yitbarek Wolde**

Dear Yitbarek Wolde,

Thank you very much. All of this will be addressed during the resubmission phase.

*This comment has been addressed as per the comment.*

Best regards,

Ashenafi Ali and co-authors.

**CC3- Sileshi W Gudeta**

Dear Sileshi W Gudeta,

Thank you very much. We have considered all comments and we are improving.

*Kindly, see sections: 2.4.3; 3.2.2; 3.2.3; 3.3; 3.4 and 4.0.*

Best regards,

Ashenafi Ali and co-authors.

**CC4- Fuat Kaya**

We thank Fuat Kaya for having an interest in the work and voluntary community
review. We respond to the key issues raised as indicated  below:

Dear Associate Editor,

I have carefully read the study As the voluntary "commentor" of the article "Reference Soil
Groups Map of Ethiopia Based on Legacy Data and Machine Learning Technique:
EthioSoilGrids 1.0".

Since I am not an official referee, my comments are sincere.

The authors should be commended for their work in Ethiopia, feeling sincerely about the data
sharing process.

**Response 1:** We are grateful for the positive comments

However, the authors have edited this article to produce only one output. I have concerns
about research questions. There are many challenges to address in digital soil mapping. And
these challenges are voiced by the DSM community. Here's an example: Ten challenges for
the future of pedometrics.

(https://www.sciencedirect.com/science/article/pii/S0016706121002354).
**Response 2:** Thank you for bringing this to our attention, we are aware of the publication you
indicated and found it helpful.

In this regard, I invite the author, who does the modeling in this valuable team, to model the
events globally with two more accepted algorithms in SoilGrids 1.0 and SoilGrids 2.0.

https://soil.copernicus.org/articles/7/217/2021/--SoilGrids 2.0: producing soil information for
the globe with quantified spatial uncertainty------Used https://journals.plos.org/plosone/article?id=10.1371/journal.pone.0105992---SoilGrids1km
— Global Soil Information Based on Automated Mapping

**Response 3:** This work considered the SoilGrids 250m (2017) as a base which succeeded the
development of the SoilGrids 1km (https://www.isric.org/explore/soilgrids/faq-soilgrids-
2017). As indicated in the Soil Grids2.0 (https://soil.copernicus.org/articles/7/217/2021/), the
numeric soil variables were only modelled and mapped (but not the soil reference groups/soil
types). We understand that SoilGrids250m (2017) is the framework in which soil type/class
modelling and mapping are done using Random Forest (RF), and as shown in lines 178 to
188 of this manuscript, RF was used for EthioGrid 1.0.

Specific comments:

Line 1:

As far as We know, This map not "conventional", well this map "digital" map.
I think "digital" must added to title.

**Response 4:** It is possible to qualify the map by adding "Digital" to the title. However, digital
maps can be generated either based on a predictive/digital soil mapping framework or
digitalised conventional maps. Therefore to avoid confusion, we prefer to qualify the map as
it is generated based on the legacy soil data and machine learning techniques which explicitly
indicate that the digital soil mapping approach was followed.

Line 35:

Really, honestly, "awesome" work for this team to collaboratively extract and collate the
data. But, We (DSM community and public) know, Soilgrids 1.0 and 2.0 versions have been
released. Publishing by running a single algorithm here is just to produce an output. There is
a need for an approach to address current DSM issues. We know that there is something
"Unknown" in Big data. And we will discover the unknown in Data with machine learning
algorithms. So why one algorithm. Comparative results are necessary for this study to make
accurate inferences for regional results.multinomial logistic regression for Soilgrids 1.0 and
quantile random forests for Soilgrids 2.0. If reference soil groups are estimated in the field
with these algorithms, their outputs will be appreciated by the DSM community at the
international level.

**Response 5:** Yes, the data extraction and compilation process is something that we are proud
of. Regarding the algorithm used as explained under response 3, the scope of the work is not
to compare algorithms, but to develop SoilGrid1.0 using a selected algorithm.

Line 70:

the last part of the introduction, the authors define a brief research purpose/question. In the last paragraph of the Introduction chapter, the Authors wrote that ... objectives of this study. In this part of the article, I rather expected a clearly formulated research goal. I suggest that in the article it is precisely stated what the purpose of the research is, using the example statement: "The goal of the study / research was ...". When formulating the research goal (s), it would be worth writing what was the cognitive (scientific) goal and what was the utilitarian (useful) goal. Before stating the purpose of the study, it would be worth formulating the research problem. The research problem may constitute a premise to indicate a gap in the current state of knowledge. It is worth writing what the current gaps in knowledge the Authors would like to fill in on the basis of planned and conducted research.

**Response 6:** Thank you for this specific comment, we will revisit and clear up confusing statements.

Line 178:

Is it just "model accuracy" ?

How do we evaluate uncertainty?

To evaluate classification-based algorithms that produce probabilistic predictions, D.G. I recommend Rossiter's valuable work.

https://www.sciencedirect.com/science/article/pii/S0016706116303901#bb0110

Please control "confusion index" released by Burroug et al. (1997 -- https://www.sciencedirect.com/science/article/pii/S0016706197000189) And the other 2 sources applied quantify in different regions, large and small areas.

https://www.sciencedirect.com/science/article/pii/S0016706116304864

https://www.tandfonline.com/doi/full/10.1080/02571862.2022.2059115

**Response 7:** The accuracy assessment (overall, user's and producer's accuracy) method and uncertainty are indicated in lines 361 to 365. Among the reviewed techniques, we have used the most commonly used cross-validation technique and accordingly the 95% confidence interval is indicated (lines 362 and 363). These are in line with the approach followed by global/regional soil grid development frameworks. However, as you indicated, there are various accuracy assessment techniques or issues that need to be considered in selecting an accuracy assessment of modelling soil classes e.g. accounting for taxonomy distance (which has also different sub-techniques), spatial cross-validation which is presumed to have limitations, dealing with clustered samples for assessing map accuracy by cross-validation, and dealing with imbalanced data in categorical mapping which might lead to issues on the accuracy of majority and minority classes. We recommend future studies to consider these issues in line 441 to 444.

Line 263:

What "reference" soil group did the models predict in areas with these classes? Is there a
taxonomic relationship here? Please read this title paper: Accounting for taxonomic distance
in accuracy assessment of soil class predictions

**Response 8:** Thank you for the recommendation. The reference soil groups indicated in line
263 were excluded from the modelling and hence comparison was not made. However, we
now get insights to include some RSGs left unmapped and improve the accuracy of this beta
version. As indicated in the confusion matrix even those soil groups modelled and mapped
have depicted different accuracy values and we noticed that some reference soil groups are
mapped at the expense of others which enables to interpret taxonomic relationships.

Line 305:

Climate, Organism and topgrapy. If it is related to them, how would it be to compile it with a
sentence?

**Response 9:** It indicates the relative importance of the predictor variables in determining the
spatial distribution of reference soil groups across the landscapes of Ethiopia. It is an effort to
go beyond prediction and incorporate model interpretations i.e. extract information on the
relationships among variables found by the models. However, as is clearly indicated in
various kinds of literature, model interpretations are not straightforward/simple in
complex/ensemble models e.g. Wadoux et al. (2022): Beyond prediction: methods for
interpreting       complex       models       of       soil       variation,
https://www.sciencedirect.com/science/article/abs/pii/S0016706122002609?via%3Dihub

Line 420, Figure 7:Very nice map. Most probable class maps, I think, for True phrase

**Response 10:** We are grateful for the appreciation.

**CC5- Sky Wills**

Dear Sky Wills (CC5),

Kindly please refer to our response to RC1; RC1 and CC5 are the same.

Kind regards,

Ashenafi Ali (on behalf of the co-authors)